# Epitopological learning and Cannistraci-Hebb network shape intelligence brain-inspired theory for ultra-sparse advantage in deep learning

**Yingtao Zhang**[1,2,3], **Jialin Zhao**[1,2,3], **Wenjing Wu**[1,2,3], **Alessandro Muscoloni**[1,2,4],
**& Carlo Vittorio Cannistraci**[1,2,3,4] [*]

[1]Center for Complex Network Intelligence (CCNI)
[2]Tsinghua Laboratory of Brain and Intelligence (THBI)
[3]Department of Computer Science, [4]Department of Biomedical Engineering
Tsinghua University, Beijing, China.

## Abstract

Sparse training (ST) aims to ameliorate deep learning by replacing fully connected artificial neural networks (ANNs) with sparse or ultra-sparse ones, such as brain networks are, therefore it might benefit to borrow brain-inspired learning paradigms from complex network intelligence theory. Here, we launch the ultra-sparse advantage challenge, whose goal is to offer evidence on the extent to which ultra-sparse (around 1% connection retained) topologies can achieve any leaning advantage against fully connected. Epitopological learning is a field of network science and complex network intelligence that studies how to implement learning on complex networks by changing the shape of their connectivity structure (epitopological plasticity). One way to implement Epitopological (epi- means new) Learning is via link prediction: predicting the likelihood of non-observed links to appear in the network. Cannistraci-Hebb learning theory inspired the CH3-L3 network automata rule for link prediction which is effective for general-purpose link prediction. Here, starting from CH3-L3 we propose Epitopological Sparse Meta-deep Learning (ESML) to apply Epitopological Learning to sparse training. In empirical experiments, we find that ESML learns ANNs with ultra-sparse hyperbolic (epi-)topology in which emerges a community layer organization that is meta-deep (meaning that each layer also has an internal depth due to power-law node hierarchy). Furthermore, we discover that ESML can in many cases automatically sparse the neurons during training (arriving even to 30% neurons left in hidden layers), this process of node dynamic removal is called percolation. Starting from this network science evidence, we design Cannistraci-Hebb training (CHT), a 4-step training methodology that puts ESML at its heart. We conduct experiments on 7 datasets and 5 network structures comparing CHT to dynamic sparse training SOTA algorithms and the fully connected counterparts. The results indicate that, with a mere 1% of links retained during training, CHT surpasses fully connected networks on VGG16, GoogLeNet, ResNet50, and ResNet152. This key finding is an evidence for ultra-sparse advantage and signs a milestone in deep learning. CHT acts akin to a gradient-free oracle that adopts CH3-L3-based epitopological learning to guide the placement of new links in the ultra-sparse network topology to facilitate sparse-weight gradient learning, and this in turn reduces the convergence time of ultra-sparse training. Finally, CHT offers the first examples of parsimony dynamic sparse training because, in many datasets, it can retain network performance by percolating and significantly reducing the node network size. Our code is available at: https://github.com/biomedical-cybernetics/Cannistraci-Hebb-training

---

[*]Corresponding author, kalokagathos.agon@gmail.com

## 1 INTRODUCTION

The human brain can learn using sparse and ultra-sparse neural network topologies and spending few watts, while current artificial neural networks are mainly based on fully connected topologies. How to make profits from ultra-sparse brain-inspired principles to ameliorate deep learning is the question that we try to address in this study, where we launch the ultra-sparse advantage challenge: the goal is to offer evidence on the extent to which ultra-sparse (around 1% connection retained) topologies can achieve any leaning advantage against fully connected. The ultra-sparse advantage challenge is in line with the current research in AI, focusing on how to reduce the training burden of the models to increase its trade-off efficiency between training speed, performance, and model size.

There exist two popular methods known as pruning (LeCun et al., 1989; Hassibi et al., 1993; Li & Louri, 2021; Sun et al., 2023; Frantar & Alistarh, 2023; Zhang et al., 2023a; Liu et al., 2023) and sparse training (Mocanu et al., 2018; Lee et al., 2019; Evci et al., 2020; Yuan et al., 2021; 2022; Zhang et al., 2022; 2023b) which introduce sparsity into deep learning. However, there are fundamental differences between them. Pruning begins with a fully connected network and gradually eliminates connections, aiming to strike a balance between post-training inference speed and final performance. Sparse training methods initiate with a sparsely connected network, resulting in inherent sparsity in both the forward and backward processes during training. Consequently, the primary goal of sparse training methods is to find the equilibrium balance between training acceleration and final performance. Dynamic Sparse Training (DST) (Mocanu et al., 2018; Evci et al., 2020; Yuan et al., 2021), a variant of sparse training, is always considered the most efficient strategy due to its ability to evolve the network's topology dynamically. However, it's worth noting that many DST algorithms don't solely rely on the intrinsic network sparsity throughout the training process. For instance, approaches like RigL (Evci et al., 2020) leverage gradient information from missing links, making the network fully connected during the backpropagation phase. Similarly, OptG (Zhang et al., 2022) introduces an extra super-mask, adding learnable parameters during training. These DST methods deviate from the core principle of sparse training, which aims to accelerate model training solely through the existing connections in the network. Integrating theories from topological network science, particularly those inspired by complex network intelligence theory designed for sparse topological scenarios, into DST could yield fascinating results.

Epitopological Learning (EL) (Daminelli et al., 2015; Durán et al., 2017; Cannistraci, 2018) is a field of network science inspired by the brain that explores how to implement learning on networks by changing the shape of their connectivity structure (epitopological plasticity) and forming epitopological engrams (memory traces). One approach to implementing EL is via link prediction, which predicts the existence likelihood of each non-observed link in a network. EL was developed alongside the Cannistraci-Hebb (CH) network automata learning theory (Daminelli et al., 2015; Durán et al., 2017; Cannistraci, 2018) according to which: the sparse local-community organization of many complex networks (such as the brain ones) is coupled to a dynamic local Hebbian learning process and contains already in its mere structure enough information to predict how the connectivity will evolve during learning partially.

The inspiration behind this theory is that for many networks associated with complex systems such as the brain, the shape of their connectivity is learned during a training process, which is the result of the dynamic evolution of the complex systems across the time (Narula, 2017; Cannistraci, 2018; Cannistraci et al., 2013a; Abdelhamid et al., 2023). This means that the evolution of the complex system carves the network structure, forming typical features of network complexity such as clustering, small-worldness, power-lawness, hyperbolic topology, and community organization (Muscoloni & Cannistraci, 2018). In turn, given a network with a shape characterized by these recognizable features of complexity, a significant (better than random) part of its future connectivity can be predicted by local network automata, which interpret the information learned in the engrams (memory traces) of the complex network topology (Narula, 2017; Cannistraci, 2018; Cannistraci et al., 2013a). Based on these notions, network shape intelligence is the intelligence displayed by any topological network automata to perform valid (significantly more than random) connectivity predictions without training by only processing the input knowledge associated with the local topological network organization (Abdelhamid et al., 2023). The network automaton training is unnecessary because it performs predictions extracting information from the network topology, which can be regarded as an associative memory trained directly from the connectivity dynamics of the complex system (Abdelhamid et al., 2023).

CH3-L3 (Muscoloni et al., 2020), one of the network automata rules under CH theory, is effective for general-purposed link prediction in bipartite networks. Building upon CH3-L3, we propose Epitopological Sparse Meta-deep Learning (ESML) to implement EL in evolving the topology of ANNs based on dynamic sparse training procedures. Finally, we design Cannistraci-Hebb training (CHT), a 4-step training methodology that puts ESML at its heart, with the aim to enhance prediction performance. Empirical experiments are conducted across 7 datasets (MNIST, Fashion_MNIST, EMNIST, CIFAR10, CIFAR100, TinyImageNet, and ImageNet2012) and 5 network structures (MLPs, VGG16, GoogLeNet, ResNet50, and ResNet152), comparing CHT to dynamic sparse training SOTA algorithms and fully connected counterparts. The results indicate that with a mere 1% of links retained during training, CHT surpasses fully connected networks on VGG16, GoogLeNet, and ResNet50. This key finding is evidence of ultra-sparse advantage and signs a milestone in deep learning. To help the reader get familiar with the concepts of network science introduced in this article, we provide a glossary that summarizes their definitions in Appendix C.

## 2 Related Work

### 2.1 Dynamic Sparse Training

The basic theoretical and mathematical notions of dynamic sparse training together with the several baseline and SOTA algorithms including SET (Mocanu et al., 2018), MEST (Yuan et al., 2021) and RigL (Evci et al., 2020) - which we adopt for comparison in this study - are described in the Appendix D. We encourage readers who are not experts on this topic to take a glance at this section in Appendix D to get familiar with the basic concepts before going forward and reading this article.

### 2.2 Epitopological Learning and Cannistraci-Hebb network automata theory for link prediction

Drawn from neurobiology, Hebbian learning was introduced in 1949 (Hebb, 1949) and can be summarized in the axiom: "neurons that fire together wire together". This could be interpreted in two ways: changing the synaptic weights (weight plasticity) and changing the shape of synaptic connectivity (Cannistraci et al., 2013a; Daminelli et al., 2015; Durán et al., 2017; Cannistraci, 2018; Narula, 2017). The latter is also called epitopological plasticity (Cannistraci et al., 2013a), because plasticity means "to change shape" and epitopological means "via a new topology". Epitopological Learning (EL) (Daminelli et al., 2015; Durán et al., 2017; Cannistraci, 2018) is derived from this second interpretation of Hebbian learning, and studies how to implement learning on networks by changing the shape of their connectivity structure. One way to implement EL is via link prediction: predicting the existence likelihood of each nonobserved link in a network. In this study, we adopt CH3-L3 (Muscoloni et al., 2020) as link predictor to regrow the new links during the DST process. CH3-L3 is one of the best and most robust performing network automata which is inside Cannistraci-Hebb (CH) theory (Muscoloni et al., 2020) that can automatically evolve the network topology with the given structure. The rationale is that, in any complex network with local-community organization, the cohort of nodes tends to be co-activated (fire together) and to learn by forming new connections between them (wire together) because they are topologically isolated in the same local community (Muscoloni et al., 2020). This minimization of the external links induces a topological isolation of the local community, which is equivalent to forming a barrier around the local community. The external barrier is fundamental to keeping and reinforcing the signaling in the local community, inducing the formation of new links that participate in epitopological learning and plasticity. As illustrated in Figure 4, CH3-L3 accomplishes this task by incorporating external local community links (eLCLs) at the denominator of a formula that ensures common neighbors minimize their interactions outside the local community. The mathematical formula of CH3-L3 and its explanation are reported in Appendix A.

## 3 Epitopological Sparse Meta-deep Learning and Cannistraci-Hebb Training

### 3.1 Epitopological Sparse Meta-deep Learning (ESML)

In this section, we propose epitopological sparse meta-deep Learning (ESML), which implements Epitopological Learning in the sparse deep learning ANN. In the standard dynamic sparse training

(DST) process, as shown in Figure 6 (left side), there are three main steps: initialization, weight update, and network evolution. Network evolution consists of both link removal and link regrowth that in standard DST is random (Erdős–Rényi model like initialization) or gradient-based Figure 6 (left side). ESML initially follows the same steps of standard DST link removal (Figure 6, left side), but then it does not implement the standard random or gradient-based link growth. Indeed, ESML progresses with substantial improvements (Figure 6, right side): new additional node and link removal part (network percolation) and new regrowth part (EL via link prediction). ESML changes the perspective of training ANNs from weights to topology. As described in Section 2.2, Epitopological Learning is a general concept detailing how the learning process works by evolving the topology of the network. This topological evolution can take various forms, such as link removal and link regrowth. In this article, we explore the benefits that link regeneration within Epitopological Learning offers to dynamic sparse training. The topological update formula in this article is:

$$\mathbf{T}^{n+1} = \mathbf{T}^n + LP_{TopK}(\mathbf{T}^n), \tag{1}$$

Where $\mathbf{T}^n$ represents the topology of the system at state $n$ and $\mathbf{T}^{n+1}$ represents the topology of the system at state $n + 1$. $LP$ can be any link predictor, such as SPM (Lü et al., 2015), CN (Newman, 2001), or CH3-L3 (Muscoloni et al., 2020), to predict the likelihood and select the TopK missing links between pair of nodes in the network. Through an internal examination, we observed that CH3-L3 outperforms SPM and CN. Once the network percolation (Figure 6, right side), which removes inactive neurons and links, is finished, Epitopological Learning via link prediction is implemented to rank the missing links in the network. This is done by computing likelihood scores with the link predictor (CH3-L3 in this article) and then regrowing the same number of links as removed with higher scores. For the necessity to contain the article's length, the details of the two removal steps are offered in Appendix G. ESML can be repeated after any backpropagation. The most used update interval in dynamic sparse training can be in the middle of an epoch, after an epoch, or after any certain number of epochs. Thus, ESML is independent of the loss function, and after ESML predicts and adds the new links to the network topology, the backpropagation only transfers the gradient information through the existing links of topology $\mathbf{T}$ in an epoch. Our networks are ultra-sparse (1% connectivity) hence the gradient information navigates to find a minimum in a smaller dimensional space with respect to the fully connected. Compared to dense, fully connected network backpropagation, in the link prediction-based dynamic sparse training, the gradient is computed considering only the existing links in the ultra-sparse topology. This means that if we preserve only 1% of the links, we only need to compute the gradients for those existing links and backpropagate them to the previous layers, which in principle could imply a 100 times training speed up.

## 3.2 Cannistraci-Hebb Training

ESML is a methodology that can work better when the network has already formed some relevant structural features. To address these requirements, we propose a novel 4-step training procedure named Cannistraci-Hebb training (CHT). **The first step is the sparse topological initialization**. For instance, the fact that in each layer, the nodes display a hierarchical position is associated with their degree. Nodes with higher degrees are network hubs in the layer, and therefore, they have a higher hierarchy. This hierarchy is facilitated by a hyperbolic network geometry with power-law-like node degree distribution (Cannistraci & Muscoloni, 2022). We propose to adopt a *Correlated sparse topological initialization (CSTI)*, which is obtained by considering links with the strongest Pearson correlations between node features in the input layer. However, in case CHT is applied to intermediate layers of a network model such as CNN, the input information is randomized by the convolutional operation applied in the previous layer and this affects the ability of the Pearson correlation to extract patterns of associations between the nodes features. Hence, the network structure can be initialized by using generative network models such as Erdős-Rényi, Watts-Strogatz, and Barabási-Albert, as we discuss in the Appendix M.

The dimension of the hidden layers is determined by a scaling factor denoted as '×'. A scaling factor of 1× implies that the hidden layer's dimension is equal to the input dimension, while a scaling factor of 2× indicates that the hidden layer's dimension is twice the input dimension, which allows the dimension of the hidden layer to be variable. In fact, since ESML can efficiently reduce the dimension of each layer, the hidden dimension can automatically reduce to the inferred size. **The second step is** *sparse weight initialization* **(SWI)**, which addresses the importance of weight initialization in the sparse network. The standard initialization methods such as Kaiming (He et al., 2015)

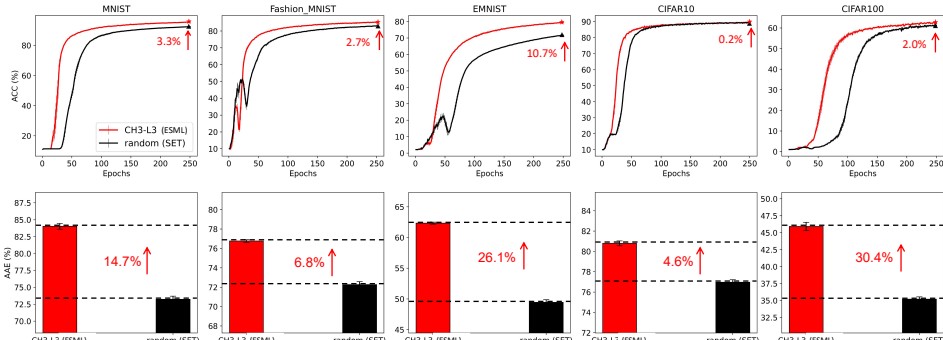

Figure 1: The comparison of CH3-L3 (ESML) and random (SET). In the first row, we report the accuracy curves of CH3-L3 (ESML) and random (SET) on 5 datasets and mark the increment at the 250th epoch on the upper right corner of each panel. The second row reports the value of area across the epochs (AAE) which indicates the learning speed of different algorithms corresponding to the upper accuracy plot. Each panel reports the percentage increase.

or Xavier (Glorot & Bengio, 2010) are designed to keep the variance of the values consistent across layers. However, these methods are not suitable for sparse network initialization since the variance is not consistent with the previous layer. To address this issue, we propose a method that can assign initial weights in any sparsity cases. In Appendix E, we provide the mathematical formula for SWI and the rationale that brought us to its definition. **The third step is the epitopological prediction**, which corresponds to ESML (introduced in Section 3.1), and it is at the 'heart' of CHT. **The fourth step is early stop and weight refinement**, during the process of epitopological prediction, it is common to observe an overlap between the links that are removed and added, as shown in the last plot in Figure 8. After several rounds of topological evolution, once the network topology has already stabilized, ESML may continuously remove and add mostly the same links, which can slow down the training process. To solve this problem, we introduce an early stop mechanism for each sandwich layer. When the overlap rate between the removal and regrown links reaches a certain threshold (we use a significant level of 90%), we stop the epitopological prediction for that layer. Once all the sandwich layers have reached the early stopping condition, the model starts to focus on learning and refining the weights using the obtained network structure. Description and ablation tests for each of these steps are in Figure Figure 5 and 8 and we detail the steps of CSTI in Appendix B.

## 4 RESULTS

Because of the page limitation, we detail our experimental setup in Appendix H.

### 4.1 RESULTS FOR ESML AND NETWORK ANALYSIS OF EPITOPOLOGICAL-BASED LINK REGROWN

#### 4.1.1 ESML PREDICTION PERFORMANCE

In this subsection, we aim to investigate the effectiveness of ESML from a network science perspective. We utilize the framework of ESML and compare CH3-L3 (the adopted topological link predictor) with randomly assigning new links (which is the regrown method of SET (Mocanu et al., 2018)). Figure 1 compares the performance of CH3-L3 (ESML) and random (SET). The first row shows the accuracy curves of each algorithm and on the upper right corner of each panel, we mark the accuracy improvement at the 250th epoch. The second row of Figure 1 shows the area across the epochs (AAE), which reflects the learning speed of each algorithm. The computation and explanation of AAE can be found in the Appendix L. Based on the results of the accuracy curve and AAE values, ESML (CH3-L3) outperforms SET (random) on all the datasets, demonstrating that the proposed epitopological learning method is effective in identifying lottery tickets(Frankle & Carbin, 2019), and the CH theory is boosting deep learning.

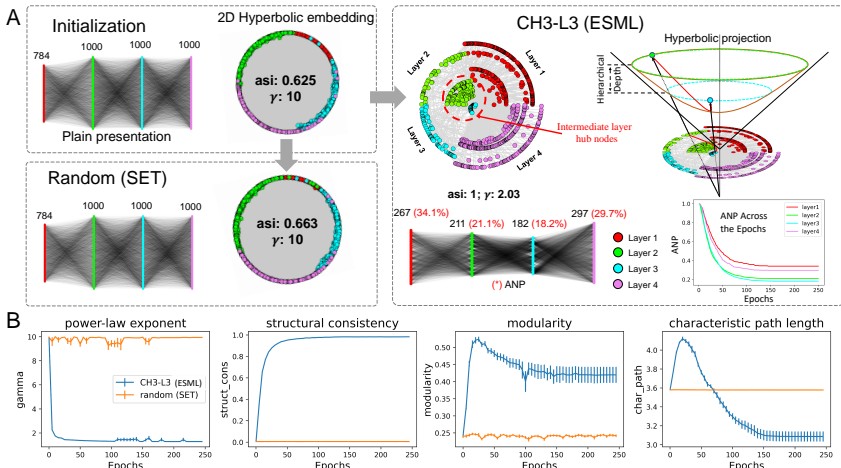

Figure 2: Network science analysis of the sparse ANNs topology. Various states of the network in **A** (a 1% density MLP trained on EMNIST) are depicted through both a plain representation and a 2D hyperbolic embedding. This includes the network randomly initialized, the network at its final epoch after Random (SET), and the network at its final epoch after CH3-L3 (ESML) training. The blocks within the network also report two key metrics: the angular separability index (ASI, which signifies the capacity to distinguish communities) and the power law exponent $\gamma$ (reflecting degree distribution). The ESML panel also displays the active neuron post-percolation rate (ANP) trend over the epochs. **B** shows the results of four network topological measures drawn from the average of five datasets by CH3-L3 (ESML) and random (SET). The shadow area represents the standard error across these five datasets.

### 4.1.2 NETWORK ANALYSIS

Figure 2 provides the analysis from the network science perspective. Figure 2A provides evidence that the sparse network trained with ESML can automatically **percolate** the network to a significantly smaller size and form a hyperbolic network with community organization. The example network structure of the EMNIST dataset on MLP is presented, where each block shows a plain representation and a hyperbolic representation of each network at the initial (that is common) and the final epoch status (that is random (SET) or CH3-L3 (ESML)).

**Network percolation.** The plain representation of the ESML block in Figure 2A shows that each layer of the network has been percolated to a small size, especially the middle two hidden layers where the active neuron post-percolation rate (ANP) has been reduced to less than 20%. This indicates that the network trained by ESML can achieve better performance with significantly fewer active neurons and that the size of the hidden layers can be automatically learned by the topological evolution of ESML. ANP is particularly significant in deep learning because neural networks often have a large number of redundant neurons that do not contribute significantly to the model's performance. Moreover, we present the analysis of the ANP reduction throughout the epochs within the ESML blocks/sandwiches. From our results, it is evident that the ANP diminishes to a significantly low level after approximately 50 epochs. This finding highlights the efficient network percolation capability of ESML. At the same time, it suggests that a prolonged network evolution may not be necessary for ESML, as the network stabilizes within a few epochs, indicating its rapid convergence.[1]

**Hyperbolic Hierarchical Organization.** Furthermore, we perform coalescent embedding (Cacciola et al., 2017) of each network in Figure 2A into the hyperbolic space, where the radial coordinates are associated with node degree hierarchical power-law-like distribution and the angular coordinates with geometrical proximity of the nodes in the latent space. This means that the typical tree-like structure of the hyperbolic network emerges if the node degree distribution is power law (gamma exponent of the power law distribution $\gamma \leq 3$ (Cannistraci & Muscoloni, 2022)). Indeed, the more

---

[1]A commented video that shows how ESML shapes and percolates the network structure across the epochs is provided at this link `https://www.youtube.com/watch?v=b5lLpOhb3BI`

power law in the network, the more the nodes migrated towards the center of the hyperbolic 2D representation, and the more the network displays a hierarchical hyperbolic structure.

**Hyperbolic Community Organization.** Surprisingly, we found that the network formed by CH3-L3 finally (at the 250th epoch) becomes a hyperbolic power law ($\gamma$=2.03) network with the hyperbolic community organization (angular separability index, $ASI = 1$, indicates a perfect angular separability of the community formed by each layer (Muscoloni & Cannistraci, 2019)), while the others (initial and random at the 250th epoch) do not display either power law ($\gamma$=10) topology with latent hyperbolic geometry or crystal-clear community organization (ASI around 0.6 denotes a substantial presence of aberration in the community organization (Muscoloni & Cannistraci, 2019)). Community organization is a fundamental mesoscale structure of real complex networks (Muscoloni & Cannistraci, 2018; Alessandro & Vittorio, 2018), such as biological and socio-economical (Xu et al., 2020). For instance, brain networks (Cacciola et al., 2017) and maritime networks (Xu et al., 2020) display distinctive community structures that are fundamental to facilitate diversified functional processing in each community separately and global sharing to integrate these functionalities between communities.

**The Emergency of Meta-depth.** ESML approach not only efficiently identifies important neurons but also learns a more complex and meta-deep network structure. With the prefix meta- (meaning after or beyond) in the expression *meta-depth*, we intend that there is a second intra-layer depth *beyond* the first well-known inter-layer depth. This mesoscale structure leverages a topologically separated layer-community to implement diversified and specialized functional processing in each of them. The result of this 'regional' layer-community processing is then globally integrated together via the hubs (nodes with higher degrees) that each layer-community owns.

Thanks to the power-law distributions, regional hubs that are part of each regional layer-community emerge as meta-deep nodes in the global network structure and promote a hierarchical organization. Indeed, in Figure 2A right we show that the community layer organization of the ESML-learned network is meta-deep, meaning that also each layer has an internal hierarchical depth due to power-law node degree hierarchy. Nodes with higher degrees in each layer community are also more central and cross-connected in the entire network topology, playing a central role (their radial coordinate is smaller) in the latent hyperbolic space, which underlies the network topology. It is as the ANN has a meta-depth that is orthogonal to the regular plan layer depth. The regular plan depth is given by the organization in layers, the meta-depth is given by the topological centrality of different nodes from different layers in the hyperbolic geometry that underlies the ANN trained by ESML.

**Network Measure Analysis.** To evaluate the structure properties of the sparse network, we utilize measures commonly used in network science (Figure 2B). We consider the entire ANN network topology and compare the average values in 5 datasets of CH3-L3 (ESML) with random (SET) using 4 network topological measures. The plot of **power-law** $\gamma$ indicates that CH3-L3 (ESML) produces networks with power-law degree distribution ($\gamma \leq 3$), whereas random (SET) does not. We want to clarify that SET (Mocanu et al., 2018) also reported that SET could achieve a power-law distribution with MLPs, but in their case, they used the learning rate of 0.01 and extended the learning epochs to 1000. We acknowledge that maybe with a higher learning rate and a much longer waiting time, SET can achieve a power-law distribution. However, we emphasize that in Figure 2B (first panel) ESML achieves power-law degree distribution regardless of conditions in 5 different datasets and 2 types (MLPs, VGG16) of network architecture. This is because ESML regrows connectivity based on the existing network topology according to a brain-network automaton rule, instead SET regrows at random. Furthermore, **structure consistency** (Lü et al., 2015) implies that CH3-L3 (ESML) makes the network more predictable, while **modularity** shows that it can form distinct obvious communities. **The characteristic path length** is computed as the average node-pairs length in the network, it is a measure associated with network small-worldness and also message passing and navigability efficiency of a network (Cannistraci & Muscoloni, 2022). Based on the above analysis, ESML-trained network topology is ultra-small-world because it is small-world with a power-law degree exponent lower than 3 (more precisely closer to 2 than 3) (Cannistraci & Muscoloni, 2022), and this means that the transfer of information or energy in the network is even more efficient than a regular small-world network. All of these measures give strong evidence that ESML is capable of transforming the original random network into a complex structured topology that is more suitable for training and potentially a valid lottery-ticket network. This transformation can lead to significant improve-

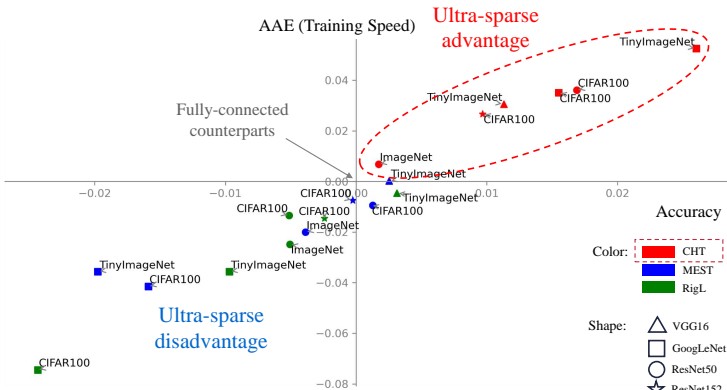

Figure 3: Percentage change in Accuracy-AAE for DST Methods retaining 1% of links compared to fully connected counterparts. Each scatter indicates the best performance of that dataset in 1× or 2× cases. In every task, CHT (red) surpasses its fully connected counterparts in both performance and training speed. This indicates that it has gained an ultra-sparse advantage.

ments in network efficiency, scalability, and performance. This, in turn, supports our rationale for introducing the novel training process, CHT.

**Credit Assigned Paths.** Epitopological sparse meta-deep learning percolates the network, triggering a hyperbolic geometry with community and hierarchical depth, which is fundamental to increase the number of credit assigned paths (CAP, which is the chain of the transformation from input to output) in the final trained structure. A detailed discussion of the emergence of CAPs in the ESML of Figure 2A is provided in Appendix O.

## 4.2 RESULTS OF CHT

**Main Results.** In Figure 3, we present the general performance of CHT, RigL, and MEST relative to the fully connected network (FC). For this comparison, the x-axis and y-axis of the left figure represent the change rate (increase or decrease) in accuracy and area across epochs (AAE, indicating learning speed) compared to the fully connected configuration. The value is computed as $(v - fc)/fc$, where $v$ stands for the measurement of the algorithms and $fc$ represents the corresponding fully connected counterparts' performance. In this scenario, algorithms positioned further to the upper right of the figure are both more accurate and faster. Table 1 provides the specific performance details corresponding to Figure 3. Observing the figure and the table, CHT (depicted in red) consistently surpasses the other dynamic sparse training methods and even the fully connected network in performance and also exhibits the fastest learning speed. Noteworthily, our method consistently retains only 1% of the links throughout the training. The preliminary results that we gained on 3 datasets are consistent and show that, in general, the static sparse training via ER, BA, or WS is outperformed by the respective dynamic sparse training that, in turn, is outperformed by CHT. In addition, in some cases, CHT initialized with BA and WS displays better performance than CHT initialized by ER or CSTI. A detailed discussion on different initialization can be found in Appendix M

**Sparsity.** Additionally, we delved into a sensitivity analysis of sparsity range from 0.999 to 0.9, depicted in Figure 7C, which reveals astonishing results. CHT exhibits good performance even at an ultra-sparse level of 0.999, whereas other dynamic sparse training methods lag behind. This strongly suggests that under the guidance of topological link prediction, dynamic sparse training can rapidly and accurately identify the optimal subnetworks, thereby improving training efficiency with minimal information.

**Running Time Analysis.** We evaluate the running times of CHT in comparison to other dynamic sparse training methods from two aspects. Firstly, we consider the area across the epochs (AAE), which reflects the learning pace of the algorithms. As emphasized in Table 1, our method consistently surpasses other dynamic sparse training algorithms in learning speed, even outpacing the dense configuration. Secondly, we assess from the standpoint of actual running time. The acceler-

Table 1: Results of CHT comparing to DST methods and fully connected network (FC) on $1\times$ and $2\times$ cases. In this table, we report the accuracy (ACC) and area across the epochs (AAE) of each case. We executed all the DST methods in an ultra-sparse scenario (1%). The best performance among the sparse methods is highlighted in bold, and values marked with "*" indicate they surpass those of the fully connected counterparts. S represents the total number of training samples within the dataset, and C denotes the number of classes present in that dataset.

| | VGG16-TinyImageNet S:100K C:200 | | GoogLeNet-CIFAR100 S:50K C:100 | | GoogLeNet-TinyImageNet S:100K C:200 | | ResNet50-CIFAR100 S:50K C:100 | | ResNet50-ImageNet S:1.2M C:1000 | | ResNet152-CIFAR100 S:50K C:100 | |
| --- | --- | --- | --- | --- | --- | --- | --- | --- | --- | --- | --- | --- |
| | ACC | AAE | ACC | AAE | ACC | AAE | ACC | AAE | ACC | AAE | ACC | AAE |
| $FC_{1\times}$ | 51.34±0.12 | 43.55±0.04 | 76.64±0.1 | 62.9±0.07 | 52.63±0.08 | 45.14±0.02 | 78.13±0.13 | 65.39±0.08 | 75.04±0.05 | 63.27±0.03 | 78.3±0.03 | 65.53±0.09 |
| $RigL_{1\times}$ | 51.32±0.25 | 43.0±0.12 | 74.12±0.34 | 56.96±0.68 | 51.23±0.12 | 42.59±0.2 | 77.34±0.04 | 64.07±0.07 | 74.5±0.02 | 60.8±0.13 | 78.08±0.08 | 64.54±0.05 |
| $MEST_{1\times}$ | 51.47±0.1* | 43.56±0.04* | 75.22±0.57 | 58.85±1.08 | 51.58±0.15 | 43.28±0.15 | 78.23±0.05* | 64.77±0.02 | 74.65±0.1 | 61.83±0.17 | 78.47±0.14* | 65.27±0.05 |
| $CHT_{1\times}$ | **51.92±0.08*** | **44.88±0.03*** | **77.52±0.10*** | **64.70±0.04*** | **53.28±0.03*** | **46.82±0.01*** | **79.45±0.02*** | **67.75±0.02*** | **75.17±0.02*** | **63.70±0.07*** | **79.2±0.16*** | **67.51±0.1*** |
| $FC_{2\times}$ | 50.82±0.05 | 43.24±0.03 | 76.76±0.21 | 63.11±0.09 | 51.46±0.13 | 44.43±0.1 | 77.59±0.03 | 65.29±0.03 | 74.91±0.02 | 63.52±0.04 | 78.49±0.07 | 65.76±0.02 |
| $RigL_{2\times}$ | 51.5±0.11* | 43.35±0.05* | 74.89±0.44 | 58.41±0.44 | 52.12±0.09* | 43.53±0.01 | 77.73±0.10* | 64.51±0.10 | 74.66±0.07 | 61.7±0.06 | 78.3±0.14 | 64.8±0.03 |
| $MEST_{2\times}$ | 51.36±0.08* | 43.67±0.05* | 75.54±0.01 | 60.49±0.02 | 51.59±0.07* | 43.25±0.03 | 78.08±0.07* | 64.67±0.11 | 74.76±0.01 | 62.0±0.03 | 78.45±0.13 | 64.98±0.16 |
| $CHT_{2\times}$ | **51.78±0.06*** | **44.19±0.12*** | **77.95±0.08*** | **65.32±0.07*** | **54.00±0.05*** | **47.51±0.08*** | **79.16±0.21*** | **67.36±0.14*** | **75.1±0.09*** | **63.43±0.18** | **79.25±0.1*** | **67.37±0.03*** |

ation benefits are derived from quicker convergence and the early-stop mechanism integrated into topological evolution. Notably, the time complexity for the link regrowth process is $O(N \times d^3)$, where $d$ represents the maximum degree of any node within the network. When the network is ultra-sparse, the time required for evolution remains relatively low. Through practical testing on a 1024-1024 network with a density of 1%, the link prediction duration is roughly 1-2 seconds. Although the runtime for the link predictor is slower than the others, we utilize an early-stop approach that truncates topological adjustments within a few epochs. As presented in Figure 10B, which exhibits the actual running times of various algorithms from initiation to convergence, CHT is consistently faster than the others. Given CHT's oracle-like capacity to intuitively suggest the placement of new links that boost gradient learning, it greatly abbreviates the convergence period for dynamic sparse training.

**Adaptive Percolation.** As depicted in Figure 10A, CHT adeptly percolates the network to preserve approximately 30-40% of the initial node size for relatively simpler tasks such as MNIST when applied with MLPs. In contrast, for more complicated tasks like ImageNet on ResNet50, the ANP remains high. This behavior underscores that CHT isn't just a myopic, greedy algorithm. It seems that it intelligently adjusts the number of active neurons based on the complexity of the task at hand, ensuring no compromise on performance, even in ultra-sparse scenarios. In contrast, other sparse training methods like RigL do show a lower ANP rate on ImageNet when using ResNet50. However, this reduction is accompanied by a performance drop, possibly because such methods lack task awareness.

This adaptive percolation ability of CHT to provide a minimalistic network modeling that preserves performance with a smaller architecture is a typical solution in line with Occam's razor also named in science as the *principle of parsimony*, which is the problem-solving principle advocating to search for explanations constructed with the smallest possible set of elements (Gauch Jr et al., 2003). On this basis, we can claim that CHT represents the first example of an algorithm for "parsimony ultra-sparse training". We stress that the parsimony is obtained as an epiphenomenon of the epitopological learning, indeed neither random nor gradient link regrowth is able to trigger the adaptive percolation of ESML or CHT.

## 5 CONCLUSION

In this paper, we investigate the design of ultra-sparse (1% links) topology for deep learning in comparison to fully connected, introducing a novel approach termed Epitopological Sparse Meta-deep Learning (ESML), which leverages insights from network science and brain-inspired learning to evolve deep artificial neural network topologies in dynamic ultra-sparse (1% links) training. Furthermore, we present the Cannistraci-Hebb Training (CHT) process, comprising four essential components to augment ESML's performance. The performance and network analysis indicate that ESML can convert the initial random network into a complex, structured topology, which is more conducive to training and potentially represents a valid lottery ticket network. Further empirical findings demonstrate that anchored by the 4-step methodology, CHT not only learns more rapidly and accurately but also adaptively percolates the network size based on task complexity. This performance surpasses other dynamic sparse training methods and even outperforms the fully connected counterparts in 6 tests by maintaining just 1% of its connections throughout the training process. This achievement marks a significant milestone in the field of deep learning. The discussion of limitations and future challenges are provided in Appendix F.

ACKNOWLEDGE

C.V.C thanks his master student professors: Prof. Giancarlo Ferrigno for inspiring his interest for sparsity in neural networks and Prof. Franco Maria Montevecchi for the support and encouragement to continue this research over the years. We thank Yuchi Liu, Yue Wu, YuanYuan Song, Yining Xin, Giada Zhou, Shao Qian Yi, Lixia Huang, and Weijie Guan for the administrative support at THBI; Hao Pang for the IT support at THBI. This Work is supported by the Zhou Yahui Chair professorship of Tsinghua University, the starting funding of the Tsinghua Laboratory of Brain and Intelligence, and the National High-level Talent Program of the Ministry of Science and Technology of China.

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

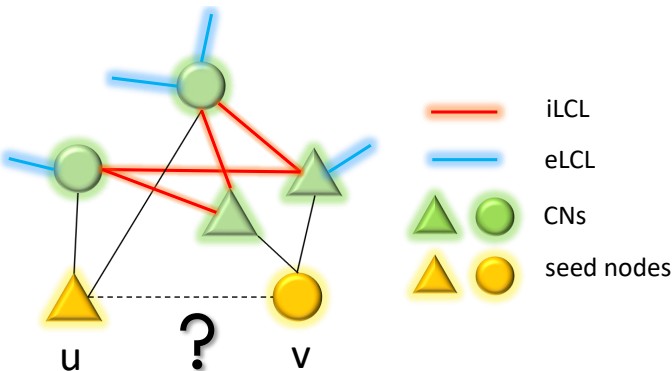

Figure 4: An explanatory example of topological link prediction performed using CH theory on bipartite networks. The two yellow nodes are the seed nodes whose non-observed interaction should be scored with a likelihood. The green nodes represent the common neighbors (CNs) of the seed nodes in the bipartite network. The red links denote the interactions between CNs, which are referred to as internal local community links (iLCL). The blue links represent the interaction between CNs and external nodes of the community that are not shown in this example which are termed as external local community links (eLCL).

## A  MATHEMATICAL FORMULA AND EXPLANATION OF CH3-L3

CH3-L3 is the link predictor we employ in the epitopological sparse meta-deep learning phrase of ESML and CHT. In this section, we delve into both the formula and underlying principles of CH3-L3 to enhance understanding of this concept.

CH3-L3 can be formalized as follows:

$$CH3 - L3(u, v) = \sum_{z_1, z_2 \in l3(u,v)} \frac{1}{\sqrt{(1 + de_{z_1}) * (1 + de_{z_2})}}, \tag{2}$$

where: $u$ and $v$ are the two seed nodes of the candidate interaction; $z_1, z_2$ are the intermediate nodes on the considered path of length three, which are seen as the common neighbors of $u$ and $v$ on the bipartite network; $de_{z_1}$, $de_{z_2}$ are the respective number of external local community links of each common neighbor; and the summation is executed over all the paths of length three (l3) between the seed nodes u and v. The formula of $CH3 - Ln$ is defined on the path of any length, which adapts to any type of network organization (Muscoloni et al., 2020). Here, we consider the length 3 paths of $CH3(CH3-L3)$ because the topological regrowth process is implemented on each ANN sandwich layer separately, which has bipartite network topology organization. And, bipartite topology is based on the path of length 3, which displays a quadratic closure (Daminelli et al., 2015). During the link prediction process, all the missing links in the network will be ranked based on scores generated by the $CH3 - L3$ formula. A higher score suggests a higher probability of the link being present in the next stage of the network.

## B  CORRELATED SPARSE TOPOLOGICAL INITIALIZATION (CSTI)

To enhance the performance of the link predictor, we proposed the Correlated Sparse Topological Initialization (CSTI) to initialize the topology of the layers that can directly interact with the input features. We explain here the detailed steps of CSTI.

As shown in Figure 5A, CSTI consists of 4 steps. 1) **Vectorization:** During the vectorization phase, we follow a specific procedure to construct a matrix of size $n \times M$, where n represents the number of samples selected from the training set. M denotes the number of valid features obtained by excluding features with zero variance ($Var_0$) among the selected samples. 2) **Feature selection:** Once we have this $n \times M$ matrix, we proceed with feature selection by calculating the Pearson Correlation for each feature. This step allows us to construct a correlation matrix. 3) **Connectivity selection:** Subsequently, we construct a sparse adjacency matrix, where the positions marked with "1" (represented as white in the heatmap plot of Figure 5A) correspond to the top-k% values from

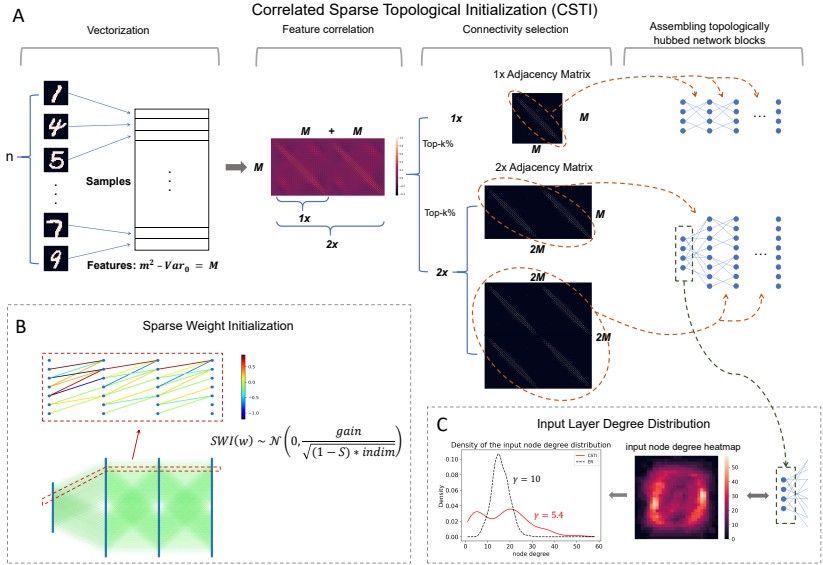

Figure 5: Main innovations introduced in CHT (CSTI and SWI). **A** shows an example of how to construct the CSTI on the MNIST dataset, which involves four steps: vectorization, feature correlation, connectivity selection, and assembling topological hubbed network blocks. **B** presents a real network example of how SWI assigns weights to the sparse network. **C** displays the input layer degree distribution along with the associated heatmap of $2\times$ case on the MNIST figure template and the node degree distribution.

the correlation matrix. The specific value of k depends on the desired sparsity level. This adjacency matrix plays a crucial role in defining the topology of each sandwich layer. The dimension of the hidden layer is determined by a scaling factor denoted as '$\times$'. A scaling factor of $1\times$ implies that the hidden layer's dimension is equal to the input dimension, while a scaling factor of $2\times$ indicates that the hidden layer's dimension is twice the input dimension, which allows the dimension of the hidden layer to be variable. In fact, since ESML can efficiently reduce the dimension of each layer, the hidden dimension can automatically reduce to the inferred size. 4) **Assembling topologically hubbed network blocks:** Finally, we implement this selected adjacency matrix to form our initialized topology for each sandwich layer.

## C    GLOSSARY OF NETWORK SCIENCE

In this section, we introduce some basic notions and concepts of network science involved in this article. Given a simple undirected network $G(V, E)$ where $V$ is the set of nodes and $E$ is the set of links. $N$ is the number of nodes in the article ($N = |V|$) and $M$ is the total number of edges in the network ($M = |E|$). A is the adjacency matrix of the network, $A_{ij} = 1$ if nodes $i$ and $j$ are connected and $A_{ij} = 0$ otherwise.

**Scale-free Network.**    A scale-free network (Barabási & Albert, 1999) is a type of node degree heterogeneous network characterized by a highly uneven distribution of connections among its nodes, where a small number of nodes (hubs) have a very high number of connections, and a large number of nodes have few connections. The degree of scale-free networks always follows power-law distribution $P(k) \sim k^{-\gamma}$, where $\gamma$ is a constant in the range of 2 to 3. In contrast, the degrees of a random network always follow a Binomial distribution (Barabási, 2013)

**Characteristic Path Length.**    The Characteristic Path Length is a fundamental network measure to quantify the average shortest distance between pairs of nodes within a network. It provides insight into how efficiently the information or signal is transferred across the network on average. The formula can be expressed as:

$$L = \frac{1}{N(N-1)} \sum_{i \neq j} d(i, j)$$

Where $d(i, j)$ is the shortest path length between node $i$ and node $j$.

**Watts–Strogatz Model and Small-World Network.** A Watts-Strogatz model of small-world network (Watts & Strogatz, 1998) is characterized by (1) a high clustering coefficient and (2) a short average path length among nodes. This short average path lengths, on the other hand, mean that there is a short chain of connections between any two nodes (in social networks, this effect is known as six degrees of separation (Budrikis, 2023)). Therefore, the Watts–Strogatz model (Watts & Strogatz, 1998) is well-known for its 'small-world' properties, characterized by high clustering and short path lengths. This model has a parameter $\beta$ and displays transitions from a regular high clustered lattice ($\beta = 0$) to a random small-world graph ($\beta = 1$). Intermediate values of $\beta$ can generate networks that retain clustering while displaying small-world connectivity, striking a balance between regularity and randomness.

More in general, the necessary condition to be a small-world network (Newman & Watts, 1999) is to have (2) a short average path length among nodes, which mathematically is defined by a network where the path length $L$ between two randomly chosen nodes grows proportionally to the logarithm of $N$ in the network, that is:

$$L \propto \log N$$

**Structural Consistency.** Structural consistency (Lü et al., 2015) is an index that is based on the first-order matrix perturbation of the adjacency matrix, which can reflect the inherent link predictability of a network and does not require any prior knowledge of the network's organization. A perturbation set $\Delta E$ is randomly selected from the original links set $E$, where $E^L$ is the set of top-$L$ ranked links given by SPM (Lü et al., 2015), $L = |\Delta E|$. The structural consistency $\sigma_c$ is computed as:

$$\sigma_c = \frac{|E^L \cap \Delta E|}{|\Delta E|}$$

**Modularity** In network science, modularity (Newman, 2006) quantifies the ability to divide a network into modules (also known as communities or clusters). The modularity of a network ranges from -1 to 1. A high modularity score (close to 1) suggests that the network exhibits dense connections between nodes within modules but sparse connections between nodes in different modules. A modularity score of 0 indicates that the network structure lacks any discernible community organization. Nodes within the network are randomly connected, and there are no clear groupings or clusters. Imagine a network where interactions between nodes are essentially uniform: no specific communities emerge. Modularity 0 implies a homogeneous network without distinct communities. When the modularity score approaches -1, it signifies an anti-community structure. Nodes are strongly interconnected across the entire network, and there is little differentiation into separate groups. Think of it as a network where everyone is closely linked to everyone else, regardless of any community boundaries. In practice, achieving modularity of exactly -1 is rare, but values close to it indicate a highly cohesive network. However, such extreme anti-community behavior is not common in real-world networks. The formula for modularity $Q$ is:

$$Q = \frac{1}{2m} \sum_{ij} \left[ A_{ij} - \frac{k_i k_j}{2m} \right] \delta(c_i, c_j)$$

Where: $A_{ij}$ represents the adjacency matrix of the network, $k_i$ and $k_j$ are the degrees of nodes $i$ and $j$, respectively. $\delta(c_i, c_j)$ is the Kronecker delta function, which equals 1 if nodes $i$ and $j$ are in the same community and 0 otherwise.

**Coalescent Embedding.** Coalescent embedding (Muscoloni et al., 2017) is a class of topological-based machine learning algorithms for nonlinear unsupervised dimensionality reduction and embedding of networks in a geometric space such as hyperbolic one. In our case, coalescent embedding is used to map networks that have latent hyperbolic geometry onto the two-dimensional hyperbolic space. The approach always involves 4 steps: 1) pre-weighting links with topological rules that approximate the underlying network geometry (Cannistraci & Muscoloni, 2022); 2) non-linear dimension reduction including three (but not limited to these) manifold-based methods: Isomap (Balasubramanian & Schwartz, 2002), non-centered Isomap (Muscoloni et al., 2017), Laplacian eigen-

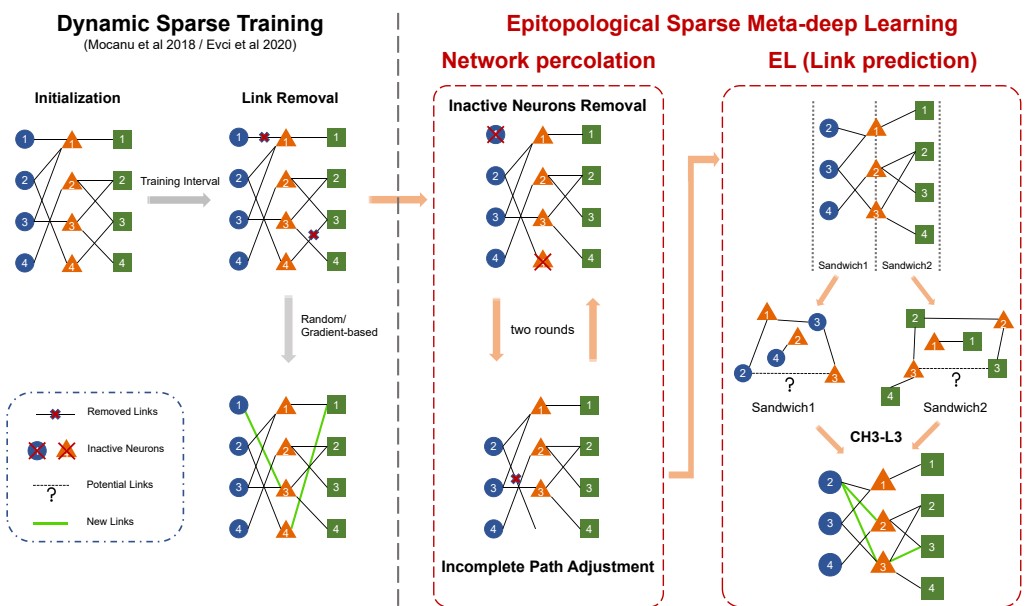

Figure 6: Illustration of the standard DST progress and ESML. ESML incorporates two additional steps for network evolution, namely network percolation and EL(link prediction). Network percolation involves Inactive Neurons Removal (INR) and the Incomplete Path Adjustment (IPA). In the epitopological learning (EL) part, which is based on link prediction, after dividing the entire network into several sandwich layers, the chosen link predictor (CH3-L3) separately calculates the link prediction score for each non-existing link and then regrows links with top-k scores in equal number to those that were removed.

maps (Belkin & Niyogi, 2003) and two minimum-curvilinearity-based methods: minimum curvilinear embedding (Cannistraci et al., 2013b) and non-centered minimum curvilinear embedding (Cannistraci et al., 2013b); 3) generate angular coordinates; 4) generate radial coordinates. In this way, we can effectively embed the original network into 2D hyperbolic space, reconstructing its latent manifold structure for visualization.

## D DYNAMIC SPARSE TRAINING

In this section, we introduce some basic concepts and relevant algorithms of dynamic sparse training. Let's first define how the network will learn through gradients with the sparse mask created by dynamic sparse training. Consider $\theta \in \mathbb{R}^N$ as the weights matrix of the entire network, $\mathbf{M} \in \{0,1\}^N$ is the adjacency matrix of the current existing links inside the model. The general optimization function can be formulated as:

$$\min_{\theta} \mathcal{L}(\theta \odot \mathbf{M}; \mathbf{D}) \quad \text{s.t.} \quad ||\mathbf{M}||_0 \leq k, \tag{3}$$

where the $\mathcal{L}(\cdot)$ is the loss function and $\mathbf{D}$ is the training dataset. $k$ is the remaining weights in the entire number $N$ of weights regarding the preset sparsity $S$, $k = N * (1 - S)$. The main difference between dynamic sparse training and static sparse training is whether it allows the mask $M$ to change dynamically. A standard dynamic sparse training method (Mocanu et al., 2018; Evci et al., 2020; Yuan et al., 2021), as illustrated in Figure 6A, begins with a sparse initialized network and undergoes the *prune-and-grow* regime with multiple iterations after each update interval of weights. In one update iteration, the gradient of the weights can be formulated as:

$$\frac{\partial \mathcal{L}}{\partial \theta_i} = \frac{\partial \mathcal{L}}{\partial (\theta_i \odot \mathbf{M}_i)} \mathbf{M}_i, \tag{4}$$

$i$ indicates the layer index of the model. The main differences between the current standard dynamic sparse training methods are how to remove links and how to regrow new links. SET (Mocanu et al., 2018), as the first dynamic sparse training method, removes links with weight magnitude and regrows new links randomly. Different from the removal process of SET, MEST (Yuan et al.,

2021) also regrows new links randomly while removing weights with the combination of weight magnitude and gradient magnitude of the existing links. RigL (Evci et al., 2020) utilizes the gradient magnitude of the missing links to regrow the new links. Some other innovations in dynamic sparse training are adding some tricks or making it more hardware-friendly. (Zhang et al., 2023b) introduces a method called Bi-Mask that accelerates the backpropagation process of sparse training with N:M constraint (Mishra et al., 2021). SpFDE (Yuan et al., 2022) introduces layer freezing and data sieving strategy to further accelerate the training speed of DST. We let you notice that Mixture-of-Experts (MoEs) (Riquelme et al., 2021) is also a type of DST method that is widely used in training large language models since the experts are always sparse during training.

However, most methods for training sparse neural networks are not actually sparse during training. For example, RigL (Evci et al., 2020) needs to collect gradients for missing connections to regrow new ones. OptG (Zhang et al., 2022) trains an extra weight mask to learn the network topology. Only, methods like SET and MEST that randomly regrow connections actually use just the sparse model during training. However, they don't work as well as methods that use gradient information. So in this paper, we introduce an efficient, gradient-free, and brain-inspired method from network science called epitopological learning to evolve the sparse network topology.

## E  SPARSE WEIGHT INITIALIZATION

Our derivation mainly follows kaiming(He et al., 2015). Similar to SNIP(Lee et al., 2019) we introduce auxiliary indicator variables $\mathbf{C} \in \{0,1\}^{m \times n}$. Then, for a densely connected layer, we have:

$$\mathbf{y}_l = \mathbf{C}_l \odot \mathbf{W}_l \mathbf{x}_l + \mathbf{b}_l, \tag{5}$$

where $\odot$ denotes the Hadamard product, $\mathbf{W}^{m \times n}$ is the weight matrix, $\mathbf{x}^{n \times 1}$ is the input vector, $\mathbf{b}^{m \times 1}$ is a vector of biases and $\mathbf{y}^{m \times 1}$ is the output vector. We use $l$ to index a layer.

Then, for $i$-th element of the pre-activation layer $\mathbf{y}_l$, by ignoring the influence of bias, we can get its variance:

$$Var[y_l^i] = Var[\sum_{j=1}^{n} c_l^{ij} w_l^{ij} x_l^j]. \tag{6}$$

As we initialize elements in $\mathbf{C}$ to be independent and identically distributed random variables, also same for elements in $\mathbf{W}$. We also assume elements in $\mathbf{x}$ are mutually independent and share the same distribution. $\mathbf{C}$, $\mathbf{W}$ and $\mathbf{x}$ are independent of each other. Variance of $y$ will be:

$$Var[y_l^i] = nVar[c_l^{ij} w_l^{ij} x_l^j] = n((Var[c] + \mu_c^2)(Var[w] + \mu_w^2)(Var[x] + \mu_x^2) - \mu_c^2 \mu_w^2 \mu_x^2), \tag{7}$$

where $\mu$ is mean. We assume $c$ follows Bernoulli distribution, then the probability mass function is:

$$f(c) = \begin{cases} S & c = 0 \\ 1 - S & c = 1 \end{cases}, \tag{8}$$

where $S$ denotes the sparsity. We could infer that $Var[c] = S(1 - S)$ and $\mu_c^2 = (1 - S)^2$. Same as previous work (He et al., 2015), we define $w$ follow zero-mean Gaussian distribution, therefore $\mu_c^2 \mu_w^2 \mu_x^2 = 0$. And when activation layer is ReLU, $Var[x_l^i] + \mu_{x_l^i}^2 = \frac{1}{2} Var[y_{l-1}^i]$. Then equation 7 could be changed to:

$$Var[y_l^i] = n(Var[c] + \mu_c^2)(Var[w] + \mu_w^2)(Var[x] + \mu_x^2) = \frac{1}{2} n(1 - S) Var[w] Var[y_{l-1}^i], \tag{9}$$

We expect the variance of signal to remain the same across layers $Var[y_l^i] = Var[y_{l-1}^i]$, then:

$$Var[w] = \frac{2}{n(1 - S)}, \tag{10}$$

We denote $indim$ as $n$, which represents the dimension of the input layer of each sandwich layer. The constant $gain$ varies depending on the activation function used. For example, when using ReLU,

we set $gain$ to $\sqrt{2}$. The resulting values of $w$ are sampled from a normal distribution, resulting in the following expression:

$$SWI(w) \sim \mathcal{N}(0, \sigma^2), \sigma = \sqrt{Var[w]} = \frac{gain}{\sqrt{(1-S) * indim}}, \tag{11}$$

## F   LIMITATIONS AND FUTURE CHALLENGES

With respect to the dynamic sparse training SOTA (RigL and MEST), the computational evidence in Figure 3 on 6 empirical tests obtained in different combinations of network architectures/datasets (VGG16 on TinyImageNet; GoogLeNet on CIFAR100 and TinyImageNet; ResNet50 on CIFAR100 and ImageNet; ResNet152 on CIFAR100), demonstrate that CHT offers a remarkable ultra-sparse (1% connectivity) advantage on the fully connected baseline, whereas the current SOTA cannot. For the sake of clarity, we acknowledge that in this study, we do not compare with the specific SOTA in the respective data or architecture types because they often include methodologies with tailored technical features that are network and data-dependent. Including these specific SOTA, would make it difficult a fair comparison across data and networks because it would largely depend on the way we adapt the ultra-sparse training strategy to the specific case. The attempt of this study is instead to evaluate whether, and in which cases, CHT can help to achieve ultra-sparse advantage in comparison to the fully connected baseline, regardless of the specific network or data considered.

A limitation of not only our proposed method CHT but of all current sparse training methods is the current difficulty in deploying such unstructured sparse training methods efficiently in hardware like GPUs or specialized AI accelerators. Some studies (Hubara et al., 2021; Zhang et al., 2023b) attempt to integrate semi-structured sparsity patterns to speed up the network, but this often comes at the cost of reduced performance. Most hardware is optimized for dense matrix multiplications, while sparse networks require more dedicated data access patterns. But this limitation could also represent a challenge to address in future studies because the gain in computational speed could be relevant considering that the backpropagation and the gradient search would be implemented with respect to an ultra-sparse structure. Another important limitation to address in the next studies is to better understand the process of node percolation and to design new strategies of node removal that allow setting a priory a wished value of node sparsity lower-bound.

The future challenges of dynamic sparse training should be in our opinion to identify novel adaptive mechanisms of epitopological parsimony learning that, for different datasets and network architectures, can adaptively learn any aspect of the network structure such as node layers, node size, and link sparsity.

## G   LINK REMOVAL AND NETWORK PERCOLATION IN ESML

We detail our link removal methods and how we implement network percolation in this section.

Firstly, in the removal phase, ESML adopts the commonly used DST removal method that prunes links with lower weights magnitude (Figure 6, left side), which are considered less useful for the model's performance. We also try an alternative removal method, which combines weight and gradient magnitude as adopted by MEST. However, this approach did not yield any performance improvement to us, at least in the tested datasets.

After the process of standard DST link removal, ESML checks and corrects the topology for the presence of inactive neurons that either do not have a connection on one side of the layer or do not have any connections on both sides of the layer. The forward and backward processes can get stuck when information passes through these inactive neurons. Moreover, if a neuron becomes inactive, no new links will be assigned to it in that network according to the formula of CH3-L3. Thus, ESML implements the Inactive Neurons Removal (INR) to eliminate those inactive neurons from the topology. In particular, in the case that inactive nodes possess connections on one layer side, we need to conduct Incomplete Path Adjustment (IPA) to remove those links. Since IPA may produce some extra inactive neurons, we execute INR and IPA for two rounds. We termed the coupled process of INR and IPA as network percolation because it is reminiscent of the percolation theory in network science, which studies the connectivity behavior of networks when a fraction of the nodes

or edges are removed (Li et al., 2021). If there are still some inactive neurons, we leave them in the network and clean up them during the next evolution.

## H  EXPERIMENTAL SETUP

In Section 4, we present experimental results and network measurement analyses for ESML and SET (Mocanu et al., 2018) across three datasets (MNIST (LeCun et al., 1998), Fashion_MNIST (Xiao et al., 2017), EMNIST (Cohen et al., 2017)) using MLPs, and two datasets (CIFAR10 (Krizhevsky, 2009), CIFAR100 (Krizhevsky, 2009)) with VGG16 (Simonyan & Zisserman, 2014). In addition, we present our comparative analysis of CHT, the 4-step training methodology, on VGG16, GoogLeNet (Szegedy et al., 2015), ResNet50 (He et al., 2016), and ResNet152 across CIFAR100, Tiny-ImageNet, and ImageNet(Russakovsky et al., 2015). CH3-L3 is chosen as the representative link predictor for ESML and CHT.

The models employing dynamic sparse training in this section adopt the final three fully connected layers (excluding the last layer) of all the convolution neural networks. It's noteworthy that the default configuration for GoogLeNet and ResNet comprises just one fully connected layer. We expanded this to three layers to allow the implementation of dynamic sparse training. All experiments are conducted using three random seeds, and a consistent set of hyperparameters is applied across all methods.

**Hidden Dimension.**  It's important to highlight that the hidden dimension settings differ between the evaluations of ESML and CHT in this article. When assessing ESML and comparing it with SET, we adopt the same network structure as outlined in SET (Mocanu et al., 2018), where the hidden dimension is set to 1000 to ensure a fair comparison with SET. For the CHT evaluation, we utilize two distinct hidden dimension settings: '1×', which corresponds to the same size as the input dimension, and '2×', which is double the size of the input dimension. All the algorithms are tested considering these two dimensions.

**Baseline Methods.**  To highlight the potential of link prediction in guiding the formation of new links, in this article, we compare ESML with SET (Mocanu et al., 2018) which serves as a foundational method in dynamic sparse training that relies on the random regrowth of new links. To highlight the advantages of CHT, we compare it with two state-of-the-art dynamic sparse training algorithms: RigL (Evci et al., 2020) and MEST (Yuan et al., 2021). Both of these algorithms follow the prune-and-regrow approach. RigL prunes links by weight magnitude and regrows them based on the gradient magnitude of missing links. Additionally, it features an adaptive $\zeta$ strategy that gradually reduces the proportion of links pruned and regrown over the training epochs. Conversely, MEST employs a combination of weight and gradient magnitudes of the existing links for link removal and subsequently regrows new links at random. Notably, MEST's "EM&S" tactic, which enables training from a denser network to a sparse one, is also implemented in this article.

**Sparsity.**  In Section 4, we maintain a sparsity level of 99% (equivalent to 1% density) for all evaluations except for the sparsity sensitivity test. The sparsity pattern adopted in this study is unstructured. This is because both pattern-based and block-based patterns impose constraints on the network's topology, potentially detrimentally affecting model performance. Nonetheless, even with unstructured sparsity, CHT facilitates a quicker training acceleration due to its enhanced learning speed and rapid convergence.

## I  SENSITIVITY TEST OF ESML AND CHT

We investigate the influence of hyperparameters which are also significant for the training of ESML and CHT.

**Learning rate.**  In particular, the learning rate (lr) is a crucial factor that controls the size of the steps taken in each mini-batch during training. In dynamic sparse training, the learning rate is even more dominant. As an example, consider a scenario where weights are assigned to new links with a value of zero. If the learning rate is set to a very small value, these weights may not be able to

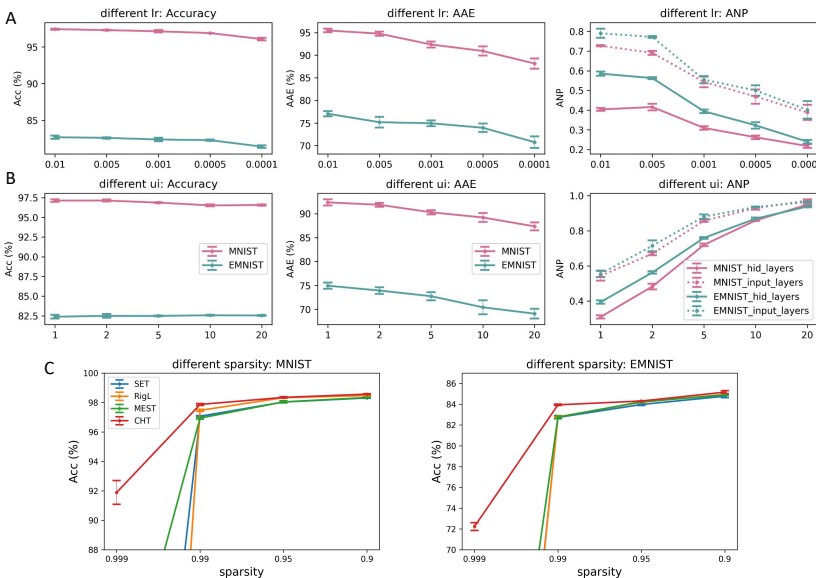

Figure 7: Sensitivity test of learning rate, update interval, and sparsity on ESML and CHT. The evaluation measures involve Accuracy, area across the epoch (AAE), and ANP (active neuron post-percolation rate). **A** shows the impact of learning rate on the MNIST and EMNIST datasets when applied to MLPs. **B** investigates how varying the update interval of topology affects performance. **C** illustrates the performance of dynamic sparse training across different sparsity levels on MNIST and EMNIST, with 1x hidden dimension ranging from 0.999 to 0.9. It's noteworthy that even at 99.9% sparsity, CHT can still learn some information, whereas the other methods cannot. All the experiments are with 3 random seeds.

increase or decrease beyond the positive or negative threshold. Consequently, the weights may get directly pruned in the next round. In Figure 7A, we fix ui=1, sparsity=0.99, and vary lr from [0.01, 0.005, 0.001, 0.0005, 0.0001]. The conclusion drawn from A and B is that a low learning rate could lead to a more percolated network with lower ANP, indicating the effectiveness of ESML in reducing the network size. However, if the learning rate is too low, the network may not be able to learn well due to the slow weight update.

**Update interval.** The update interval (ui) encounters a similar challenge. If the interval is not appropriately set, it can lead to suboptimal outcomes. The choice of update interval impacts how frequently the network structure is adjusted, and an unsuitable value may hinder the network from effectively adapting to new information. The results show that a larger ui could make the network learn slower and result in a less percolated network.

**Sparsity.** In our evaluation, we also place significant emphasis on the role of sparsity when assessing the performance of dynamic sparse training methods. Our objective is to delve into the capabilities of these methods in scenarios of extreme sparsity. In Figure 7, we keep ui=1 and lr=0.01 as constants while we manipulate the sparsity levels within the range of [0.999, 0.99, 0.95, 0.9]. In this setup, we present a comparative analysis of CHT against SET, MEST, and RigL. The purpose is to emphasize the differences among these dynamic sparse training methods as they operate under varying levels of sparsity. Analyzing the results on MNIST and EMNIST, we observed a remarkable trend: even in scenarios with an extreme sparsity level of 99.9% (density=0.1%), where the connectivity is highly sparse, CHT manages to extract and transfer meaningful information. In contrast, both RigL and SET failed to learn anything significant. This disparity in performance can be attributed to the capabilities of link prediction methods in handling sparse information. CHT's ability to utilize topology for predicting new connections becomes especially advantageous in situations of extreme sparsity. On the other hand, SET and MEST, which randomly regenerate new connections, face a vast search space, making it exceedingly challenging to identify the correct subnetwork. Meanwhile, RigL struggles in such scenarios due to the limited availability of credit assignment paths (CAPs) in the extremely sparse connectivity, rendering gradient information less useful.

**Number of Hidden Layers** In the experiments detailed in Section 4, we consistently utilize three sparse layers followed by a single fully connected layer in each model configuration. Additionally, we explore configurations with just one sparse layer and one fully connected layer, as presented in Table 2. This is compared to the original architecture of ResNet152 (only one fully connected layer). The results demonstrate that the CHT outperforms both fully connected alternatives and other DST methods. Moreover, CHT also exceeds the performance benchmarks set by the original ResNet152 configuration.

Table 2: Results of CHT comparing to DST methods and fully connected network (FC) on $1\times$ and $2\times$ cases with only 1 sparse layer. In this table, we report the accuracy (ACC) and area across the epochs (AAE) of ResNet152 on CIFAR100. We executed all the DST methods in an ultra-sparse scenario (1%). The best performance among the sparse methods is highlighted in bold, and values marked with "*" indicate they surpass those of the fully connected counterparts.

| | ResNet152-CIFAR100 S:50K C:100 | |
|---|---|---|
| | **ACC** | **AAE** |
| **Original** | 79.76±0.19 | 68.26±0.11 |
| **FC$_{1\times}$** | 79.28±0.16 | 67.33±0.21 |
| **RigL$_{1\times}$** | 79.81±0.09* | 68.07±0.09 |
| **MEST$_{1\times}$** | 79.97±0.11* | 68.42±0.05* |
| **CHT$_{1\times}$(WS, $\beta = 1$)** | **80.01±0.02*** | **68.91±0.09*** |
| **FC$_{2\times}$** | 79.22±0.12 | 67.69±0.14 |
| **RigL$_{2\times}$** | 79.59±0.03 | 67.95±0.1 |
| **MEST$_{2\times}$** | 79.86±0.08* | 68.28±0.07* |
| **CHT$_{2\times}$(WS, $\beta = 1$)** | **80.15±0.15*** | **68.9±0.13*** |

## J CHT RESULTS ON THE MLP TASKS

In the main text, we showcase the results of CHT on larger datasets and models, while here, we detail CHT's performance on basic datasets and models. From the results in Table 3, CHT surpasses other dynamic sparse training methods in accuracy on MNIST (MLP) and EMNIST (MLP) while achieving similar performance as others on Fashion MNIST. It's noteworthy that while certain dynamic sparse training methods might slightly surpass CHT in some instances, CHT consistently exhibits higher AAEs across all tasks, indicating a faster learning pace.

## K ABLATION TEST OF EACH COMPONENT IN CHT

In this section, we evaluate each component introduced in CHT (Sparse Topological Initialization, Sparse Weight Initialization, and early stop).

**Sparse Topological Initialization.** In this article, we discuss 4 types of initializations: Correlated Sparse Topological Initialization (CSTI), Erdős-Rényi (ER), Barabási-Albert, and Watts–Strogatz (WS). Note that the WS model is equivalent to the ER model if $\beta$ equals 1. For the layers that directly receive information from the inputs, we adopt Correlated Sparse Topological Initialization (CSTI). We first report the initialization structure in Figure 8C which is a CSTI example with a $2\times$ version of the hidden dimension on MNIST. We construct the heatmap of the input layer degrees and node distribution density curve. It can be observed that with CSTI, the links are assigned to nodes mostly associated with input pixels in the center of figures, and indeed, the area at the center of the figures is more informative. We also report the CSTI-formed network has an initialized power law exponent $\gamma = 5.4$, which indicates a topology with more hub nodes than in a random network with $\gamma = 10$. In Figure 8, we demonstrate that with CSTI, ESML can achieve faster learning speed compared to ER initialization, and in Table 3, we compare CSTI with 3 types of random initialization: BA, WS, and ER (WS, $\beta = 1$). The results that we gained on 3 datasets are consistent and show that, in general, the static sparse training via ER, BA, or WS is outperformed by the respective dynamic sparse training that, in turn, is outperformed by CHT. In addition, on the Fashion MNIST dataset, CHT initialized with WS and varying $\beta$ values display better performance than CHT initialized by CSTI.

For the layers in the mediate positions of the models, we compare 3 distinct models of random initialization: ER, BA, and WS in Table 4. We conduct the ablation test on these different topological initializations on CNNs. The findings indicate that the CHT achieves optimal performance on the WS network when $\beta$ varies between 0.25 and 1.0, depending on the specific case. Nevertheless,

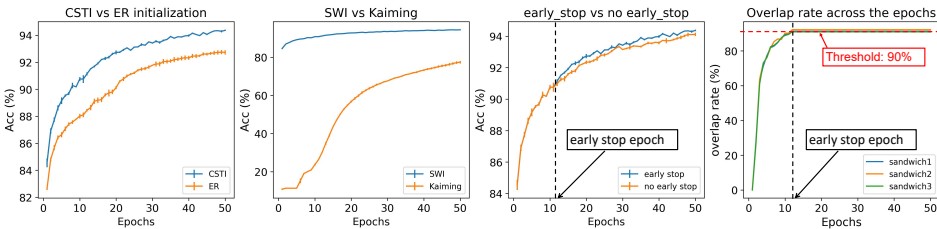

Figure 8: Ablation test of each component in CHT on the MLP architecture for MNIST dataset with 1× hidden dimension. The first three figures demonstrate the efficiency of different strategies while keeping the other components fixed and only varying the one being compared. The last figure illustrates the overlap rate of the new links and removed links, providing insight into the necessity of utilizing an early stop of the evolutionary process.

selecting the most suitable initialization for each scenario remains a topic for further exploration. For the sake of simplicity, ER (WS, $\beta = 1$) initialization is utilized as the default approach in the experiments presented in the main text.

**Sparse Weight Initialization (SWI).** On the second panel of Figure 8, we assess SWI against the commonly used weight initialization method in dynamic sparse training, Kaiming. The results indicate that SWI facilitates faster training from the initial phases.

**Early stop.** The third panel of Figure 8 presents evidence that early stop can prevent the over-evolution of the network topology. Once the network structure becomes stable, running ESML is unnecessary, and it is more optimal to conduct an early stop and focus on refining the weights to achieve convergence. As an example, at the 12th epoch, the overlap rate (the overlap between removed links and new links) exceeds 90%, which can be seen as a topological stability threshold, indicating that the network should no longer evolve. Based on this observation, we implement an early stop for the evolutionary process. With early stop, we not only achieve performance improvements but also accelerate training speed since there's no need for further topological exploration.

## L  AREA ACROSS THE EPOCHS

The Area Across the Epochs (AAE) is computed as the cumulative sum of the performance indicator of an algorithm till a certain epoch is divided by the number of epochs. Therefore, AAE is the average performance that an algorithm displays till a certain epoch during training, it is bounded [0,1] and it is an indicator of the learning speed of an algorithm. This means that algorithms whose training curves grow faster have larger AAE and can be considered: "faster learners". For instance, considering the same algorithm and fixing the same number of epochs for each trial, AAE can be used to measure and compare the impact that the tuning of a hyperparameter has on learning speed and to quantify the extent to which this hyperparameter governs learning speed.

$$AAE(A) = \frac{\int m \, dm}{E - 1},$$ (12)

Where $A$ denotes the algorithms that need to be evaluated, $E$ is the number of epochs. $m$ denotes the accuracy at each epoch.

Pay attention, when we come to comparing different algorithms, there are two important remarks about the way to apply AAE for a fair comparison of learning speed.

First, AAE should be applied considering for all different algorithms the same number of epochs and the same hyperparameter settings that can influence learning speed. Suppose an algorithm stops at a number of epochs that is smaller than others. In that case, all the algorithms' AAEs will be computed till this smallest epoch value to make sure that the algorithms are fairly evaluated considering the same interval of epochs. Suppose two algorithms are run using different update intervals of epochs during training. In that case, the comparison is not fair, and both algorithms should be run using the same update intervals. This is valid for any setting that can influence the learning speed. All

learning speed settings should be fairly fixed to the same value across methods to ensure that the learning speed difference between algorithms is intrinsic to the sparse training mechanisms and it is not biased by confounding factors.

Second, AAE can only be used to compare the learning speed of algorithms inside the same computational experiment and not across different datasets and network architectures because different datasets or network architectures can generate different learning speeds. However, if the mission of the comparison is to investigate the different learning speeds of the same algorithm with the same hyperparameter settings and network structure but using different datasets, then using AAE is still valid and can help determine how the same algorithm might react differently in learning speed because of some variations in the data. An interesting test to try is, for instance, investigating the impact of different increasing noise levels on the same type of data considering the same algorithm and setting. This would help answer the question of how noise impacts an algorithm's learning speed.

## M   SELECTING NETWORK INITIALIZATION METHODS: A NETWORK SCIENCE PERSPECTIVE

In our study, we initialize our network employing Correlated Sparse Topological Initialization (CSTI) on layers that directly interact with input features from the dataset and apply Erdős-Rényi (ER) to intermediate layers. Additionally, to facilitate a thorough comparison, we also introduce two well-known network models: the Watts–Strogatz (WS) model and the Barabási-Albert (BA) model.

The Watts–Strogatz model (Watts & Strogatz, 1998) is well-known for its 'small-world' properties, characterized by high clustering and short path lengths. This model displays transitions from a regular high-clustered lattice ($\beta = 0$) to a random small-world graph ($\beta = 1$). Intermediate values of $\beta$ can generate networks that retain clustering while displaying small-world connectivity, striking a balance between regularity and randomness. On the other hand, the Barabási-Albert model (Barabási & Albert, 1999) is noted for its scale-free nature, featuring a power-law degree distribution. This means a few nodes (hubs) have a very high degree, while most nodes have a low degree, mimicking the structure of many real-world networks. A significant limitation of both the WS and BA models is their lack of a bipartite structure, which is crucial in many real-world applications. To address this, we provide a detailed method for initializing networks using the WS and BA models in a bipartite network context. This involves adapting these models to suit the bipartite framework, thus expanding their applicability to more diverse and complex network scenarios.

**Modelling Barabási–Albert Bipartite Networks.**   1. Generate a Barabási-Albert (BA) monopartite model consisting of $n + m$ nodes, where $n$ represents the size of the first layer and $m$ the size of the second layer. 2. Randomly select $n$ and $m$ nodes in the network and assign them to their respective layers. 3. Identify and count the 'frustrations' in both layers. A frustration in layer 1 (F1) is defined as a link between nodes within the same layer. Similarly, count the frustrations in the second layer, F2. If the number of frustrations in F1 equals that in F2 ($\#F1 = \#F2$), proceed to match each frustration in F1 with one in F2 randomly. For each matched pair, apply a single rewiring step according to the Maslov-Sneppen randomization (MS) algorithm. If $\#F1 < \#F2$, adjust the process so that $\#F1$ equals the size of a randomly sampled subset of F2 ($F2s$). For the remaining frustrations in F2 ($F2r$), where $\#F2r = \#F2$–$\#F1$, sequentially break each frustration in the layer and rewire the links of each of the two nodes to the opposite layer using the preferential attachment formula from step 1. If $\#F1 > \#F2$, apply a similar procedure but adjusted to this specific case. The resulting BA network from this algorithm will be bipartite and exhibit a power-law distribution with a $\gamma$ value of 2.76. This is because the steps in this algorithm retain the same distributions as the original monopartite BA model generated in step 1.

**Modelling Watts-Strogatz Bipartite Networks.**   1) Construct a regular ring lattice with $N = \#L1 + \#L2$ nodes, where $L1$ and $L2$ represent the nodes in the first and second layers, respectively. Label the $N$ nodes proportionally such that if $\#L1 > \#L2$, then $L2$ nodes are placed every round($\#L1/\#L2$) nodes. For example, if $\#L1 = 100$ and $\#L2 = 10$, then the $L2$ nodes will be positioned in the ring every 10 $L1$ nodes. 2) For every $L1$ node, establish connections only with the

$K/2$ nearest $L2$ neighbors in the regular ring lattice. 3) For every node $i = 0, \ldots, N-1$ take every edge connecting it to its $K/2$ rightmost neighbors, that is every edge $(i, j)$ such that $0 < (j - i)$ mod $N \leq K/2$, and rewire it with probability $\beta$. Rewiring is done by replacing $(i, j)$ with $(i, k)$, where $k$ is chosen uniformly at random from all possible nodes while avoiding self-loops ($k \neq i$) and link duplication (there is no edge $(i, k')$ with $k' = k$ at this point in the algorithm). Note that when beta=1, the generated WS network is equal to ER.

**The Dynamic Version of BA and WS Network.** Following the same network evolutionary prune-and-regrow regime, we introduce the different link regrown methods based on the different network properties of BA and WS.

1) BA network: After removing a certain number of links, shooting from random nodes the same number of links using preferential attachment probability. Note that Feng et al. (2022) also adopt the preference attachment to regrow new links in the continual learning field.

2) WS network: After removing a certain number of links, shooting from random nodes the same number of links using uniformly at random probability.

This means that the BA network will always keep a scale-free network while the node degree distribution in the WS network is more uniform all the time.

**Keeping The Sparsity Fixed For All The Methods.** Since the initialization of BA and WS rely on the average degree m, which may not be integer. Therefore, the sparsity of the generated network might not be the same as we preset. We propose the below strategy to solve this problem. Given a certain m, approximate its value to the first decimal, then take the first decimal of m and call it FD. Then, a sequence of 10 nodes will be created, for which 10-FD nodes have degree [m] and FD nodes have degree [m] +1.

# N    SUBRANKING STRATEGY OF CH3-L3

Here, we describe the sub-ranking strategy adopted by CH3-L3(Muscoloni et al., 2020) to internally rank all the node pairs that have the same CH score. Although we didn't apply this strategy in the current article due to the ultra-sparse scenario, it can be useful for lower-sparsity DST applications in the future. The sub-ranking strategy aims to provide a more precise ranking for links with identical link prediction scores, particularly those that are tied-ranked and located just above the regrown threshold links.

To implement the sub-ranking strategy, the following algorithmic steps are followed:

- Assign a weight to each link $(i, j)$ in the network, denoted as $w_{i,j} = \frac{1}{1+CH_{i,j}}$.

- Compute the shortest paths (SP) between all pairs of nodes in the weighted network.

- For each node pair $(i, j)$, compute the Spearman's rank correlation ($SPcorr$) between the vectors of all shortest paths from node $i$ and node $j$.

- Generate a final ranking where node pairs are initially ranked based on the CH score ($CH_{i,j}$), and ties are sub-ranked based on the $SPcorr_{i,j}$ value. Random selection is used to rank the node pairs if there are still tied scores.

While the $SPcorr$ can be replaced with any other link predictor, we adopted this strategy because it aligns with the neurobiological principles underlying the CH model(Muscoloni et al., 2020). According to the interpretation of Peters' rule, the probability of two neurons being connected is influenced by the spatial proximity of their respective axonal and dendritic arbors(Rees et al., 2017). In other words, connectivity depends on the geometric proximity of neurons. This neurobiological concept is consistent with the $SPcorr$ score. It fits within the framework of CH modeling, as a high correlation suggests that two nodes have similar shortest paths to other nodes in the network, indicating spatial proximity due to their close geometric positioning.

Table 3: The results of sparse training on MLP models. We explored different initialization methods of CHT among Correlated Sparse Topological Initialziaiton (CSTI), Barabási-Albert (BA) network, and Watts-Strogatz (WS) network with $\beta = 0, 0.25, 0.5, 0.75, 1.0$ (Note that when $\beta = 1$, WS network is equivalent to Erdős-Rényi model). We also compare CHT with the fixed BA network and WS network with different $\beta$ and their dynamic version. In the meantime, we compare our method with the SOTA dynamic sparse training methods (RigL, MEST) and the dense network. The best performance among the sparse training methods is highlighted in bold.

| | MNIST S:60K C:10 | | Fashion_MNIST S:60K C:10 | | EMNIST S:131K C:47 | |
|---|---|---|---|---|---|---|
| | ACC | AAE | ACC | AAE | ACC | AAE |
| $\textbf{FC}_{1\times}$ | 98.69±0.02 | 97.01±0.12 | 90.43±0.09 | 88.54±0.02 | 85.58±0.06 | 83.16±0.09 |
| $\textbf{RigL}_{1\times}$ | 97.40±0.07 | 94.71±0.07 | 88.02±0.10 | 86.13±0.12 | 82.96±0.04 | 79.88±0.06 |
| $\textbf{MEST}_{1\times}$ | 97.31±0.05 | 94.41±0.05 | 88.13±0.10 | 86.02±0.07 | 83.05±0.04 | 80.02±0.07 |
| $\textbf{CHT}_{1\times}\textbf{(CSTI)}$ | **98.05±0.04** | **96.9±0.06** | 88.07±0.11 | 86.12±0.05 | **83.82±0.04** | **81.05±0.17** |
| $\textbf{CHT}_{1\times}\textbf{(BA)}$ | 96.36±0.04 | 95.79±0.03 | 87.11±0.06 | 86.28±0.11 | 80.5±0.11 | 79.14±0.07 |
| $\textbf{CHT}_{1\times}\textbf{(WS,}\ \beta = 0\textbf{)}$ | 96.08±0.05 | 95.56±0.06 | 88.27±0.14 | 87.29±0.11 | 78.8±0.16 | 77.74±0.07 |
| $\textbf{CHT}_{1\times}\textbf{(WS,}\ \beta = 0.25\textbf{)}$ | 97.18±0.03 | 96.59±0.05 | **88.5±0.02** | **87.38±0.06** | 81.12±0.04 | 80.09±0.04 |
| $\textbf{CHT}_{1\times}\textbf{(WS,}\ \beta = 0.5\textbf{)}$ | 97.25±0.02 | 96.6±0.03 | 88.19±0.04 | 87.14±0.06 | 81.33±0.06 | 80.15±0.05 |
| $\textbf{CHT}_{1\times}\textbf{(WS,}\ \beta = 0.75\textbf{)}$ | 97.06±0.05 | 96.49±0.05 | 87.84±0.13 | 86.95±0.06 | 80.97±0.07 | 79.77±0.05 |
| $\textbf{CHT}_{1\times}\textbf{(WS,}\ \beta = 1\textbf{)}$ | 96.92±0.02 | 96.37±0.03 | 87.72±0.12 | 86.83±0.12 | 81.03±0.13 | 79.71±0.15 |
| $\textbf{BA}_{1\times}$ **static** | 96.19±0.09 | 94.3±0.16 | 87.04±0.07 | 85.01±0.12 | 80.93±0.08 | 75.75±0.01 |
| $\textbf{BA}_{1\times}$ **dynamic** | 97.11±0.08 | 95.47±0.09 | 87.98±0.04 | 85.78±0.09 | 82.6±0.08 | 78.14±0.02 |
| $\textbf{WS}_{1\times}$ **static(**$\beta = 0$**)** | 95.41±0.01 | 92.01±0.10 | 86.74±0.03 | 81.71±0.11 | 77.91±0.07 | 71.06±0.21 |
| $\textbf{WS}_{1\times}$ **static(**$\beta = 0.25$**)** | 96.4±0.03 | 92.6±0.10 | 87.27±0.12 | 83±0.06 | 81.3±0.07 | 73.18±0.13 |
| $\textbf{WS}_{1\times}$ **static(**$\beta = 0.5$**)** | 96.33±0.04 | 92.13±0.09 | 87.18±0.07 | 82.49±0.01 | 81.37±0.02 | 72.45±0.03 |
| $\textbf{WS}_{1\times}$ **static(**$\beta = 0.75$**)** | 96.07±0.08 | 91.55±0.07 | 86.78±0.10 | 82.4±0.06 | 81.39±0.05 | 72.00±0.15 |
| $\textbf{WS}_{1\times}$ **static(**$\beta = 1$**)** | 96.18±0.04 | 91.43±0.08 | 86.59±0.02 | 82.53±0.07 | 81.1±0.11 | 71.75±0.12 |
| $\textbf{WS}_{1\times}$ **dynamic(**$\beta = 0$**)** | 97.04±0.04 | 94.15±0.14 | 88.09±0.05 | 84.3±0.10 | 82.79±0.09 | 75.89±0.11 |
| $\textbf{WS}_{1\times}$ **dynamic(**$\beta = 0.25$**)** | 96.95±0.04 | 93.87±0.06 | 87.85±0.14 | 84.4±0.04 | 82.42±0.08 | 75.1±0.15 |
| $\textbf{WS}_{1\times}$ **dynamic(**$\beta = 0.5$**)** | 96.88±0.05 | 93.02±0.08 | 87.78±0.09 | 83.87±0.04 | 82.45±0.03 | 74.28±0.12 |
| $\textbf{WS}_{1\times}$ **dynamic(**$\beta = 0.75$**)** | 96.94±0.06 | 92.74±0.12 | 87.57±0.07 | 83.72±0.12 | 82.55±0.06 | 74.11±0.03 |
| $\textbf{WS}_{1\times}$ **dynamic(**$\beta = 1$**)** | 96.81±0.04 | 92.56±0.07 | 87.46±0.17 | 83.91±0.08 | 82.52±0.07 | 73.92±0.08 |
| $\textbf{FC}_{2\times}$ | 98.73±0.02 | 97.14±0.03 | 90.74±0.13 | 88.37±0.05 | 85.85±0.05 | 84.17±0.11 |
| $\textbf{RigL}_{2\times}$ | 97.91±0.09 | 95.17±0.03 | 88.66±0.07 | 86.14±0.07 | 83.44±0.09 | 81.41±0.25 |
| $\textbf{MEST}_{2\times}$ | 97.66±0.03 | 95.63±0.09 | 88.33±0.10 | 85.01±0.07 | 83.50±0.09 | 80.77±0.03 |
| $\textbf{CHT}_{2\times}\textbf{(CSTI)}$ | **98.34±0.08** | **97.51±0.07** | 88.66±0.07 | 87.63±0.19 | **85.43±0.10** | **83.91±0.11** |
| $\textbf{CHT}_{2\times}\textbf{(BA)}$ | 97.08±0.05 | 96.67±0.03 | 87.48±0.05 | 86.8±0.05 | 80.38±0.04 | 79.46±0.03 |
| $\textbf{CHT}_{2\times}\textbf{(WS,}\ \beta = 0\textbf{)}$ | 96.31±0.06 | 95.89±0.01 | 87.48±0.16 | 87.83±0.07 | 79.15±0.03 | 78.28±0.03 |
| $\textbf{CHT}_{2\times}\textbf{(WS,}\ \beta = 0.25\textbf{)}$ | 97.64±0.02 | 97.29±0.06 | 88.57±0.08 | **88.21±0.13** | 81.91±0.01 | 81.02±0.01 |
| $\textbf{CHT}_{2\times}\textbf{(WS,}\ \beta = 0.5\textbf{)}$ | 97.75±0.04 | 97.36±0.04 | **88.98±0.07** | 87.73±0.08 | 81.77±0.09 | 80.77±0.09 |
| $\textbf{CHT}_{2\times}\textbf{(WS,}\ \beta = 0.75\textbf{)}$ | 97.47±0.04 | 97.01±0.05 | 88.61±0.09 | 87.42±0.04 | 81.13±0.07 | 80.37±0.10 |
| $\textbf{CHT}_{2\times}\textbf{(WS,}\ \beta = 1\textbf{)}$ | 97.39±0.05 | 97.03±0.03 | 88.17±0.11 | 87.4±0.06 | 81±0.18 | 80.22±0.03 |
| $\textbf{BA}_{2\times}$ **static** | 96.79±0.04 | 95.67±0.04 | 87.69±0.03 | 86.19±0.05 | 82.02±0.20 | 79.18±0.25 |
| $\textbf{BA}_{2\times}$ **dynamic** | 97.49±0.02 | 96.67±0.06 | 88.3±0.13 | 86.7±0.14 | 83.18±0.04 | 80.5±0.26 |
| $\textbf{WS}_{2\times}$ **static(**$\beta = 0$**)** | 96.03±0.10 | 93.41±0.14 | 87.57±0.01 | 83.72±0.28 | 78.72±0.15 | 73.09±0.20 |
| $\textbf{WS}_{2\times}$ **static(**$\beta = 0.25$**)** | 96.95±0.10 | 93.98±0.06 | 88.26±0.12 | 84.15±0.22 | 82.42±0.09 | 76.03±0.08 |
| $\textbf{WS}_{2\times}$ **static(**$\beta = 0.5$**)** | 97.27±0.07 | 93.48±0.09 | 88.26±0.08 | 83.87±0.06 | 82.66±0.08 | 75.38±0.08 |
| $\textbf{WS}_{2\times}$ **static(**$\beta = 0.75$**)** | 96.77±0.07 | 92.85±0.15 | 88.13±0.07 | 83.47±0.10 | 82.39±0.07 | 74.64±0.22 |
| $\textbf{WS}_{2\times}$ **static(**$\beta = 1$**)** | 96.67±0.06 | 92.52±0.09 | 87.68±0.04 | 83.46±0.07 | 82.29±0.18 | 74.41±0.16 |
| $\textbf{WS}_{2\times}$ **dynamic(**$\beta = 0$**)** | 97.57±0.06 | 95.46±0.19 | 88.42±0.19 | 85.39±0.25 | 83.41±0.06 | 78.6±0.20 |
| $\textbf{WS}_{2\times}$ **dynamic(**$\beta = 0.25$**)** | 97.36±0.02 | 94.72±0.06 | 88.49±0.17 | 85.35±0.14 | 83.41±0.05 | 77.46±0.08 |
| $\textbf{WS}_{2\times}$ **dynamic(**$\beta = 0.5$**)** | 97.15±0.03 | 93.82±0.11 | 88.49±0.08 | 84.71±0.05 | 83.09±0.01 | 76.26±0.08 |
| $\textbf{WS}_{2\times}$ **dynamic(**$\beta = 0.75$**)** | 97.17±0.06 | 93.77±0.19 | 88.14±0.06 | 84.56±0.09 | 82.94±0.04 | 75.94±0.17 |
| $\textbf{WS}_{2\times}$ **dynamic(**$\beta = 1$**)** | 97.14±0.01 | 93.54±0.05 | 88.11±0.07 | 84.46±0.11 | 83.14±0.07 | 75.55±0.06 |

Table 4: The results of sparse training on CNN models. We explored different initialization methods of CHT among Barabási-Albert (BA) network, and Watts-Strogatz (WS) network with $\beta = 0, 0.25, 0.5, 0.75, 1.0$ (Note that when $\beta = 1$, WS network is equivalent to Erdős-Rényi model). We also compare CHT with the fixed BA network and WS network with different $\beta$ and their dynamic version. In the meantime, we compare our method with the SOTA dynamic sparse training methods (RigL, MEST) and the dense network. The best performance among the sparse training methods is highlighted in bold, and the bolded values marked with "*" indicate that they surpass those of the fully connected counterparts.

| | VGG16-TinyImageNet S:100K C:200 | | GoogLeNet-CIFAR100 S:50K C:100 | | GoogLeNet-TinyImageNet S:100K C:200 | | ResNet50-CIFAR100 S:50K C:100 | | ResNet50-ImageNet S:1.2M C:1000 | | ResNet50-CIFAR100 S:50K C:100 | |
| | ACC | AAE | ACC | AAE | ACC | AAE | ACC | AAE | ACC | AAE | ACC | AAE |
|---|---|---|---|---|---|---|---|---|---|---|---|---|
| FC$_{1\times}$ | 51.34±0.12 | 43.55±0.04 | 76.64±0.1 | 62.9±0.07 | 52.63±0.08 | 45.14±0.02 | 78.13±0.13 | 65.39±0.08 | 75.04±0.05 | 63.27±0.03 | 78.3±0.03 | 65.53±0.09 |
| RigL$_{1\times}$ | 51.32±0.25 | 43.0±0.12 | 74.12±0.34 | 56.96±0.68 | 51.23±0.12 | 42.59±0.2 | 77.34±0.04 | 64.07±0.07 | 74.5±0.02 | 60.8±0.13 | 78.08±0.08 | 64.54±0.05 |
| MEST$_{1\times}$ | 51.47±0.1 | 43.56±0.04 | 75.22±0.57 | 58.85±1.08 | 51.58±0.15 | 43.28±0.15 | 78.23±0.05 | 64.77±0.02 | 74.65±0.1 | 61.83±0.17 | 78.47±0.14 | 65.27±0.05 |
| CHT$_{1\times}$(BA) | 51.17±0.05 | 43.35±0.01 | 77.89±0.05 | **65.32±0.08*** | 52.63±0.24 | 45.23±0.44 | 79.33±0.12 | 67.55±0.11 | 74.74±0.12 | 62.44±0.26 | 79.03±0.13 | 67.05±0.06 |
| CHT$_{1\times}$(WS, $\beta=0$) | 51.32±0.09 | 43.44±0.02 | 77.67±0.09 | 65.08±0.10 | 52.25±0.53 | 44.97±0.89 | 79.45±0.03 | 68.11±0.06 | 75.08±0.05 | **63.70±0.01*** | 79.61±0.05 | 68.15±0.15 |
| CHT$_{1\times}$(WS, $\beta=0.25$) | 51.26±0.04 | 43.57±0.04 | 77.78±0.11 | 65.04±0.07 | 51.52±0.48 | 43.22±0.89 | **79.76±0.09*** | **68.14±0.06*** | 75.12±0.2 | 63.61±0.02 | 79.21±0.06 | 67.94±0.08 |
| CHT$_{1\times}$(WS, $\beta=0.5$) | 51.32±0.05 | 43.43±0.04 | **77.95±0.03** | 65.18±0.06 | 52.46±0.44 | 44.92±0.89 | 79.14±0.04 | 67.67±0.03 | 75.08±0.10 | 63.43±0.08 | **79.74±0.02*** | **68.24±0.12*** |
| CHT$_{1\times}$(WS, $\beta=0.75$) | 51.75±0.05 | **44.94±0.01*** | 77.89±0.06 | 65.05±0.05 | **53.41±0.10*** | **46.88±0.06*** | 79.29±0.04 | 67.50±0.04 | 75.01±0.02 | 63.48±0.07 | 79.35±0.07 | 67.55±0.03 |
| CHT$_{1\times}$(WS, $\beta=1$) | **51.92±0.08*** | 44.88±0.03 | 77.52±0.10 | 64.70±0.04 | 53.28±0.03 | 46.82±0.01 | 79.45±0.02 | 67.75±0.02 | **75.17±0.02*** | 63.70±0.07 | 79.2±0.16 | 67.51±0.1 |
| BA$_{1\times}$ static | 51.10±0.02 | 43.36±0.03 | 76.31±0.07 | 61.46±0.11 | 51.43±0.09 | 43.02±0.08 | 78.40±0.08 | 64.61±0.10 | 74.75±0.05 | 62.07±0.04 | 78.32±0.14 | 65.07±0.11 |
| WS$_{1\times}$ static ($\beta=0$) | 51.27±0.04 | 43.56±0.08 | 75.62±0.14 | 59.99±0.30 | 49.77±0.10 | 40.65±0.06 | 78.38±0.10 | 64.41±0.12 | 73.96±0.07 | 60.46±0.05 | 78.51±0.00 | 64.77±0.15 |
| WS$_{1\times}$ static ($\beta=0.25$) | 51.45±0.04 | 43.74±0.02 | 75.78±0.13 | 60.15±0.19 | 50.33±0.09 | 41.31±0.02 | 77.94±0.11 | 64.40±0.09 | 74.78±0.02 | 61.52±0.08 | 78.88±0.15 | 64.74±0.05 |
| WS$_{1\times}$ static ($\beta=0.5$) | 51.42±0.03 | 43.48±0.05 | 75.58±0.15 | 60.11±0.08 | 50.00±0.05 | 40.91±0.13 | 78.45±0.03 | 64.34±0.06 | 74.33±0.04 | 61.11±0.02 | 78.62±0.05 | 64.74±0.20 |
| WS$_{1\times}$ static ($\beta=0.75$) | 51.26±0.14 | 43.27±0.04 | 74.48±0.00 | 57.23±0.00 | 49.83±0.08 | 40.62±0.13 | 78.06±0.11 | 63.54±0.20 | 74.26±0.08 | 61.10±0.09 | 78.23±0.05 | 64.86±0.05 |
| WS$_{1\times}$ static ($\beta=1$) | 51.19±0.07 | 43.14±0.06 | 75.05±0.12 | 57.62±0.18 | 49.53±0.10 | 40.63±0.03 | 77.70±0.15 | 63.60±0.10 | 74.26±0.03 | 60.84±0.02 | 78.25±0.01 | 64.69±0.14 |
| BA$_{1\times}$ dynamic | 51.21±0.05 | 43.25±0.03 | 76.41±0.15 | 61.49±0.12 | 51.73±0.11 | 42.97±0.05 | 78.25±0.05 | 64.72±0.07 | 74.42±0.04 | 61.25±0.05 | 78.86±0.10 | 65.22±0.15 |
| WS$_{1\times}$ dynamic ($\beta=0$) | 51.26±0.08 | 43.43±0.05 | 74.88±0.28 | 58.30±0.50 | 50.19±0.12 | 40.93±0.18 | 78.28±0.07 | 64.64±0.12 | 74.51±0.06 | 61.56±0.03 | 78.42±0.03 | 65.31±0.12 |
| WS$_{1\times}$ dynamic ($\beta=0.25$) | 51.30±0.04 | 43.37±0.03 | 74.52±0.16 | 57.38±0.36 | 50.58±0.06 | 41.54±0.04 | 78.11±0.06 | 64.59±0.08 | 74.68±0.05 | 61.52±0.04 | 78.17±0.10 | 64.66±0.10 |
| WS$_{1\times}$ dynamic ($\beta=0.5$) | 51.20±0.03 | 43.09±0.03 | 74.30±0.25 | 57.04±0.59 | 50.64±0.06 | 41.23±0.06 | 78.20±0.13 | 64.43±0.08 | 74.56±0.08 | 61.71±0.02 | 78.53±0.05 | 64.62±0.06 |
| WS$_{1\times}$ dynamic ($\beta=0.75$) | 51.15±0.07 | 43.42±0.03 | 74.82±1.26 | 57.06±2.72 | 51.19±0.08 | 42.50±0.08 | 77.62±0.15 | 63.25±0.17 | 74.51±0.04 | 61.49±0.01 | 78.44±0.08 | 64.28±0.11 |
| WS$_{1\times}$ dynamic ($\beta=1$) | 51.10±0.11 | 43.37±0.04 | 74.67±0.25 | 57.01±0.36 | 51.09±0.09 | 42.52±0.12 | 77.93±0.16 | 63.88±0.12 | 74.39±0.03 | 61.43±0.05 | 78.51±0.15 | 64.19±0.17 |
| FC$_{2\times}$ | 50.82±0.05 | 43.24±0.03 | 76.76±0.21 | 63.11±0.09 | 51.46±0.13 | 44.43±0.1 | 77.59±0.03 | 65.29±0.03 | 74.91±0.02 | 63.52±0.04 | 78.49±0.07 | 65.76±0.02 |
| RigL$_{2\times}$ | 51.5±0.11 | 43.35±0.05 | 74.89±0.44 | 58.41±0.44 | 52.12±0.09 | 43.53±0.01 | 77.73±0.10 | 64.51±0.10 | 74.66±0.07 | 61.7±0.06 | 78.3±0.14 | 64.8±0.03 |
| MEST$_{2\times}$ | 51.36±0.08 | 43.67±0.05 | 75.54±0.01 | 60.49±0.02 | 51.59±0.07 | 43.25±0.03 | 78.08±0.07 | 64.67±0.11 | 74.76±0.01 | 62.0±0.03 | 78.45±0.13 | 64.98±0.16 |
| CHT$_{2\times}$(BA) | 50.56±0.04 | 41.48±0.14 | 77.84±0.04 | 65.28±0.01 | 51.50±0.01 | 44.64±0.12 | 78.54±0.12 | 66.88±0.10 | 74.39±0.04 | 61.65±0.09 | 79.47±0.06 | 68.17±0.05 |
| CHT$_{2\times}$(WS, $\beta=0$) | **51.93±0.01*** | 43.92±0.01 | 77.50±0.06 | 65.15±0.05 | 52.63±0.56 | 45.37±0.93 | 79.10±0.08 | 67.81±0.05 | 74.86±0.03 | 63.17±0.03 | 79.52±0.10 | **68.10±0.08*** |
| CHT$_{2\times}$(WS, $\beta=0.25$) | 51.80±0.01 | 44.00±0.13 | 77.67±0.05 | 64.86±0.03 | 53.91±0.11 | 46.33±0.44 | **79.88±0.02*** | **68.28±0.03*** | **75.12±0.05*** | **63.55±0.09*** | **79.55±0.07*** | 67.97±0.05 |
| CHT$_{2\times}$(WS, $\beta=0.5$) | 51.64±0.05 | 44.14±0.10 | 77.59±0.03 | 65.09±0.03 | 53.24±0.50 | 45.60±0.93 | 79.07±0.02 | 67.61±0.03 | 75.04±0.03 | 63.19±0.05 | 79.44±0.05 | 67.75±0.06 |
| CHT$_{2\times}$(WS, $\beta=0.75$) | 51.74±0.09 | 44.07±0.07 | 77.53±0.04 | 64.38±0.07 | 53.85±0.10 | 47.32±0.08 | 79.10±0.03 | 67.22±0.02 | 74.83±0.02 | 62.86±0.04 | 79.39±0.04 | 67.67±0.05 |
| CHT$_{2\times}$(WS, $\beta=1$) | 51.78±0.06 | **44.19±0.12*** | **77.95±0.08*** | **65.32±0.07*** | **54.00±0.05*** | **47.51±0.08*** | 79.16±0.21 | 67.36±0.14 | 75.1±0.09 | 63.43±0.18 | 79.25±0.1 | 67.37±0.03 |
| BA$_{2\times}$ static | 51.31±0.09 | 43.52±0.00 | 76.11±0.04 | 61.33±0.10 | 51.96±0.04 | 43.58±0.03 | 78.32±0.12 | 65.07±0.09 | 74.58±0.03 | 62.08±0.01 | 78.32±0.12 | 65.07±0.09 |
| WS$_{2\times}$ static ($\beta=0$) | 51.34±0.08 | 43.92±0.03 | 76.05±0.05 | 60.89±0.03 | 50.32±0.13 | 41.47±0.10 | 78.17±0.16 | 64.57±0.17 | 74.65±0.03 | 61.67±0.08 | 78.39±0.02 | 65.11±0.08 |
| WS$_{2\times}$ static ($\beta=0.25$) | 51.32±0.05 | 43.97±0.05 | 76.01±0.05 | 60.27±0.03 | 50.95±0.12 | 42.06±0.10 | 78.02±0.08 | 64.76±0.11 | 74.62±0.03 | 61.7±0.02 | 78.65±0.07 | 65.20±0.04 |
| WS$_{2\times}$ static ($\beta=0.5$) | 51.34±0.08 | 43.69±0.03 | 75.86±0.14 | 60.26±0.15 | 50.94±0.12 | 41.92±0.10 | 77.91±0.02 | 64.57±0.06 | 74.55±0.03 | 61.57±0.03 | 78.58±0.06 | 65.02±0.05 |
| WS$_{2\times}$ static ($\beta=0.75$) | 51.47±0.07 | 43.62±0.07 | 75.26±0.18 | 59.05±0.09 | 50.62±0.07 | 41.76±0.05 | 77.81±0.05 | 64.18±0.05 | 74.58±0.02 | 61.62±0.05 | 78.57±0.04 | 64.98±0.04 |
| WS$_{2\times}$ static ($\beta=1$) | 51.31±0.05 | 43.52±0.06 | 75.28±0.12 | 58.71±0.00 | 50.57±0.16 | 41.65±0.23 | 78.07±0.09 | 64.21±0.17 | 74.49±0.02 | 61.59±0.08 | 78.56±0.03 | 64.92±0.03 |
| BA$_{2\times}$ dynamic | 51.42±0.04 | 43.36±0.04 | 76.71±0.01 | 61.70±0.03 | 51.93±0.08 | 43.18±0.06 | 78.21±0.20 | 65.02±0.12 | 74.62±0.11 | 62.02±0.04 | 78.71±0.22 | 65.02±0.15 |
| WS$_{2\times}$ dynamic ($\beta=0$) | 51.63±0.01 | 43.75±0.04 | 75.43±0.10 | 59.29±0.04 | 51.19±0.10 | 42.11±0.10 | 78.43±0.02 | 64.99±0.10 | 74.75±0.07 | 61.82±0.03 | 78.91±0.04 | 65.45±0.12 |
| WS$_{2\times}$ dynamic ($\beta=0.25$) | 51.34±0.06 | 43.57±0.05 | 74.98±0.13 | 59.28±0.17 | 51.24±0.07 | 41.94±0.09 | 77.95±0.08 | 64.66±0.08 | 74.52±0.04 | 62.48±0.09 | 78.74±0.10 | 65.62±0.10 |
| WS$_{2\times}$ dynamic ($\beta=0.5$) | 51.47±0.04 | 43.48±0.01 | 75.33±0.10 | 58.58±0.22 | 50.86±0.15 | 41.80±0.16 | 78.03±0.03 | 64.62±0.04 | 74.49±0.03 | 61.37±0.13 | 78.62±0.05 | 65.22±0.06 |
| WS$_{2\times}$ dynamic ($\beta=0.75$) | 51.22±0.05 | 43.55±0.02 | 74.43±0.25 | 57.30±0.80 | 51.26±0.06 | 42.66±0.08 | 78.19±0.06 | 64.28±0.09 | 74.51±0.03 | 61.47±0.04 | 78.45±0.08 | 64.98±0.11 |
| WS$_{2\times}$ dynamic ($\beta=1$) | 51.24±0.04 | 43.48±0.03 | 74.60±0.06 | 58.16±0.11 | 50.98±0.03 | 42.40±0.13 | 78.04±0.13 | 64.19±0.14 | 74.55±0.03 | 61.64±0.02 | 78.42±0.15 | 64.69±0.17 |

## O  CREDIT ASSIGNED PATH

The epitopological learning implemented via CH3-L3 link prediction produces a topology that encourages new inter-layer links between pair of non-interacting nodes that are topologically close to each other because they share many externally isolated (minimum number of external links with respect to the local community) common neighbors on paths of length 3 (see Figure 2). This means that during epitopological-based dynamic sparse training, the new predicted inter-layer interactions will connect with continuous path nodes with increasing higher degrees across layers, creating a meta-deep structure inside each layer that connects across the layers and that is composed of the nodes that are at the center of the hyperbolic disk because they have a higher degree and therefore they are more central in the hyperbolic geometry. Hence, epitopological sparse meta-deep learning percolates the network, triggering a hyperbolic geometry with community and hierarchical depth, which is fundamental to increase the number of credit assigned paths (CAP, which is the chain of the transformation from input to output) in the final trained structure. The total number of CAPs in the ESML network is 9,095,753, which is much larger (around 10 times) than the CAPs of SET-random, which is 782,032. Even more interestingly, we introduce a new measure for the evaluation of sparse network topology arrangement: the ratio of CAPs that pass by the hub nodes in intermediate layers. In ESML network of Figure 2A, the hubs are the 80 nodes at the center of hyperbolic disk in layer 2 and 3) and the hub-CAPs ratio is 3551460/9095753=0.39, whereas when taking the top 80 nodes with higher degree in the layer 2 and 3 of SET-random network of Figure 2A, the hub-CAPs ratio passing by them is 5596/782032=0.007. This explains how crucial epitopological learning via CH3-L3 link prediction to trigger a layer hierarchical community organized hyperbolic structure at the center of which emerges a cohort of hubs that create a frame that connects across the layers of the network, forming a scale-free topology with power-law distribution $\gamma$=2.03. The meta-deep structure of these hubs across the layers, which is associated with the hierarchy (because nodes are in the center of the hyperbolic representation) in hyperbolic geometry and to the power-law distribution triggers an ultra-small-world network with higher navigability (Cannistraci & Muscoloni, 2022).

Table 5: The detailed information of warmup and learning rate decay strategy applied in this article.

| Model | Dataset | Epochs | Starting Learning-rate | Milestones | Gamma | Batch-size | Warm-up | Warm-up Epoch |
|-------|---------|--------|-----------------------|------------|-------|-----------|---------|---------------|
| VGG16 | Tiny-ImageNet | 200 | 0.1 | 61,121,161 | 0.2 | 256 | No | - |
| GoogLeNet | CIFAR100 | 200 | 0.1 | 60,120,160 | 0.2 | 128 | Yes | 1 |
| GoogLeNet | Tiny-ImageNet | 200 | 0.1 | 61,121,161 | 0.2 | 256 | No | - |
| ResNet50 | CIFAR100 | 200 | 0.1 | 60,120,160 | 0.2 | 128 | Yes | 1 |
| ResNet152 | CIFAR100 | 200 | 0.1 | 60,120,160 | 0.2 | 128 | Yes | 1 |
| ResNet50 | ImageNet | 90 | 0.1 | 32, 62 | 0.1 | 256 | No | - |

| Method | Connection Removal | Connection Regrown | Gradient Backpropagation | Fixed Sparsity | Weight Initialization | Weight Update | Topological Initialization | Topological Early Stop |
|--------|--------------------|--------------------|--------------------------|----------------|-----------------------|---------------|----------------------------|------------------------|
| SET, DSR | Weight Magnitude | Random | Sparse | ✔ | Kaiming | zero | ER | ✘ |
| RigL | Weight Magnitude | Gradient | Dense | ✔ | Kaiming | zero | ER, ERK | ✘ |
| MEST | Weight and Gradient Magnitude | Random | Sparse | ✘ | Kaiming | zero | ER | ✘ |
| GraNet | Weight Magnitude | Gradient | Sparse | ✘ | Kaiming | zero | Dense | ✘ |
| OptG | Trainable super mask | | Dense | ✘ | Kaiming | - | ER | ✘ |
| CHT (Ours) | Weight Magnitude | **ESML** | Sparse | ✔ | **SWI** | **SWI** | **STI** | ✔ |

Figure 9: Comparing various components of different pruning and sparse training algorithms. The innovations in this article are highlighted in bold. Innovations introduced in this study, including 'ESML' (Epitopological Sparse Meta-deep Learning), 'SWI' (Sparse Weight Initialization), and 'STI' (Sparse Topological Initialization, are highlighted in bold. We also introduce an early stop mechanism for topological exploration once the network topology stabilizes.

## P  HYPERPARAMETER SETTING AND IMPLEMENTATION DETAILS

To improve the reproducibility of this paper, we have detailed the hyperparameter setting for each section here.

### P.1  HYPERPARAMETER SETTING

For the overall hyperparameter setting, we set a default sparsity=0.99 (except for the sensitivity test), the fraction of removed links zeta=0.3, weight decay=1e-05, batch size=32, and momentum =0.9 for all the MLP and VGG16 tasks. For the learning rate, we use a fixed rate for all tasks, and to highlight the distinction between ESML and SET, we set a lower learning rate of 0.0001 in Section 4.1. The detailed hyperparameter settings for comparing CHT and the other algorithms are provided in Table 5. All the experiments in this article involve image augmentation, which includes random cropping and horizontal flipping, followed by a standard normalization process. The models VGG16, GoogLeNet, ResNet50, and ResNet152 adhere to their default hyperparameter configurations, with the sole modification being the expansion of the fully connected section at the model's end to encompass 4 layers. The hidden dimension is set to either $1\times$ or $2\times$ of the input dimension.

### P.2  ALGORITHM TABLES OF CHT

Please refer to the Algorithm 1.

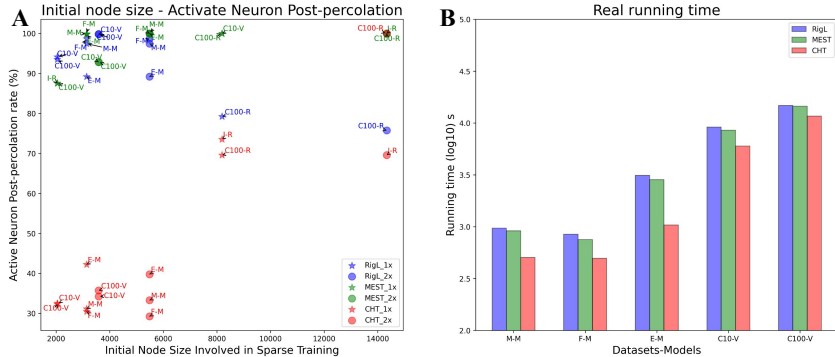

Figure 10: Active Neuron Post-percolation rate and real running time of the dynamic sparse training methods. **A** shows the incremental or reduction in Active Neuron Post-percolation corresponds to the initial node size. For datasets: 'M', 'F', and 'E' stand for MNIST, Fashion_MNIST, and EMNIST, respectively; 'C10' and 'C100' denote 'CIFAR10' and 'CIFAR100'; while 'I' signifies ImageNet2012. Regarding the models, 'M', 'V', and 'R' correspond to MLP, VGG16, and ResNet50, respectively. **B** shows the real running time (log10) of each 'Dataset-Model' task. This running time is computed from starting training to model convergence of $1\times$ case.

---

**Algorithm 1** Algorihtm of CHT.

---

**Input:** Network $f_\Theta$, dataset $\mathcal{D}$, Update Interval $\Delta t$, Early Stop Signal $\lambda$, Learning rate $\alpha$, Overlap Threshold $\sigma$.
$T \leftarrow$ Sparse Topological Initialization,
$w \leftarrow$ Sparse Weight Initialization,
$\lambda \leftarrow$ False,
**for** each training step $t$ **do**
    Sample a batch $\mathcal{B}_t \sim \mathcal{D}$,
    $\mathcal{L}_t = \sum_{i \sim \mathcal{B}_t} \mathcal{L}(f(x_i, w, T), y_i)$,
    $w = w - \alpha \nabla w \mathcal{L}_t$,
    **if** $t \mod \Delta t == 0$ and $!\lambda$ **then**
        allLayersPassed $\leftarrow$ True
        **for** each layer $l$ **do**
            $w^l, T_{t+1}^l = ESML(w^l, T_t^l)$
            allLayersPassed $\leftarrow$ allLayersPassed $\wedge$ (OverlapRate$(T_{t+1}^l, T_t^l) > \sigma$)
        **end for**
        **if** allLayersPassed **then**
            $\lambda \leftarrow$ True
        **end if**
    **end if**
**end for**

---

