# OpenReview forum: "Epitopological learning and Cannistraci-Hebb network shape intelligence brain-inspired theory for ultra-sparse advantage in deep learning"
_ICLR.cc/2024/Conference — ICLR 2024 poster_

### Official Review · Reviewer_nVAm · 2023-10-28

**Soundness:** 3 good
**Presentation:** 3 good
**Contribution:** 3 good
**Rating:** 8
**Confidence:** 2

**Summary:**

The paper introduces sparse training in deep learning, aiming to replace fully connected neural networks with ultra-sparse ones inspired by brain networks. They propose Epitopological Sparse Meta-deep Learning (ESML) using Cannistraci-Hebb learning theory. ESML learns ultra-sparse hyperbolic topologies, showcasing meta-deep organization. Empirical experiments reveal ESML's ability to automatically sparse neurons through percolation. Cannistraci-Hebb training (CHT) is introduced and compared with fully connected networks on VGG16 and ResNet50. CHT acts as a gradient-free oracle, guiding link placement for sparse-weight gradient learning. It introduces parsimony dynamic sparse training by retaining performance through percolation and reducing node network size.

**Strengths:**

- The paper is praised for its density and detail.
- The methods employed in the paper bring unique perspectives to sparse training.
- The paper is commended for conducting various experiments, with promising results.
- The inclusion of detailed experiment descriptions is noted as a positive aspect, providing transparency and allowing readers to understand the methodology.
- The inclusion of visual illustrations is a positive aspect, aiding in the comprehension of dense presentation.

**Weaknesses:**

- Simplifying the language could enhance accessibility to a wider audience.
- Some results lack standard deviation reporting, raising questions about the reliability and robustness of those specific findings.
- Addressing the legibility issue by enlarging text within figures would enhance the overall effectiveness of these visuals.

**Questions:**

The proposed method appears effective in scenarios where neural networks are overparameterized. Does it maintain its efficacy when applied to non-overparameterized neural networks characterized by limited depth and width? Particularly, how does it perform in relation to a large dataset?

---

> ### Author Response · Authors · 2023-11-20
> **Reply to the Reviewer 1203**
>
> Dear Reviewer 1203,
>
> Thank you for your detailed review and insightful questions. We also thank your suggestions to improve the paper.
>
> Below are the replies to your questions.
>
> **Weaknesses:**
>
> **1. Simplifying the language could enhance accessibility to a wider audience.**
>
> **Reply:** Because of the time limitation, we will continue to prepare a more simplified version of the language after this rebuttal. In addition, These are the actions that we implement or we intend to implement:
> + We already simplified the message in the title of the article, proposing a new title.
> + A commented video that shows how ESML shapes and percolates the network structure across the epochs for the example of Figure 2A is provided at this link  https://shorturl.at/blGY1
> + we are planning to release in the final version of the article an Appendix with a glossary of the definition of the main new concepts such as: epitopological learning, network shape intelligence, Cannistraci-Hebb Network automata, meta-deep architecture, hyperbolic network topology, community network organization, etc… To this aim, we introduced in the new version of the Introduction section the following sentence: << To help the reader to get familiar with the concepts introduced in this article, we provide a glossary that summarizes their definitions in Appendix.>>
>
> **2. Some results lack standard deviation reporting, raising questions about the reliability and robustness of those specific findings.**
>
> **Reply:** Thanks. To address the reviewer’s concern, we added the standard deviation of all the experiments.
>
> **3. Addressing the legibility issue by enlarging text within figures would enhance the overall effectiveness of these visuals.**
>
> **Reply:** Thanks, to address the reviewer’s concern, we revised the figure and enlarged text within figures in the newly uploaded version of the manuscripts.
>
> **Questions:**
>
> The proposed method appears effective in scenarios where neural networks are overparameterized. Does it maintain its efficacy when applied to non-overparameterized neural networks characterized by limited depth and width? Particularly, how does it perform in relation to a large dataset?
>
> **Reply:** Thanks to the Reviewer for this insightful comment. The largest dataset adopted in the previous version of the article is ImageNet2012 which is now still the biggest and comprehensive dataset in the computer vision field. However, to address the Reviewer's concerns we considered larger and more difficult datasets such as TinyImageNet instead of CIFAR10 on VGG16, for a more direct comparison with a new considered CNN model that is GoogLeNet. In general, the results show that CHT gains an ultra-sparse advantage on both VGG16 and GoogLeNet when applied to TinyImageNet.
>
> Finally, to investigate whether CHT maintains its efficacy when applied to non-overparameterized neural networks characterized by limited depth and width, we tested a scenario in which we applied CHT to CIFAR100 (for reason of time limitations we considered only this large dataset to attain results before rebuttal deadline) with ReseNet50 architecture whose intermediate layers depth was reduced from 4 to 2 (which is the minimum we can use).
>
> The results of this test are provided in Appendix K in Table 2 and show that CHT performance is increased from around 78% to 80% of accuracy and the ultra-sparse network advantage is retained.

---

> > ### Comment · Reviewer_nVAm · 2023-11-23
> >
> > I appreciate your prompt attention to my concerns in this revision. The authors have addressed the majority of those concerns in this revision. As a result, I've adjusted my score to reflect the positive changes made by the authors.

---

### Official Review · Reviewer_cMYk · 2023-10-31

**Soundness:** 3 good
**Presentation:** 3 good
**Contribution:** 3 good
**Rating:** 6
**Confidence:** 4

**Summary:**

This paper is about a new training methodology for deep learning called Cannistraci-Hebb training (CHT). CHT is a training methodology that uses epitopological sparse meta-deep learning (ESML) to learn artificial neural networks (ANNs) with ultra-sparse hyperbolic topology.

**Strengths:**

1. The paper is well-written and all the methodological details are described clearly.
2. CHT is a new 4-step training methodology for deep learning that uses ESML to learn ANNs with ultra-sparse hyperbolic topology.
3. CHT has been shown to surpass fully connected networks on VGG16 and ResNet50.
4. CHT could be used in a wide range of real-world applications, such as image classification, natural language processing, speech recognition, recommender systems, and medical diagnosis.

**Weaknesses:**

1. For the node structure, this paper only compares CHT with fully connected graph, the author should take other graph structure into consideration, like ER, BA, WS graph.
2. Compared with SOTA models, the performance of CHT is not competitive.

**Questions:**

1. What's the inspiration of current architecture?
2. Did you do extra experience on other graph structure?
3. In Table 2 on page15. Why some accuracy result is not the best but still highlighted?

---

> ### Author Response · Authors · 2023-11-20
> **Reply to the Reviewer cMYk**
>
> Dear Reviewer cMYk,
>
> Thank you for your review of our paper. We appreciate your positive evaluation of the soundness, presentation, and contribution of our work, as well as the specific strengths you've highlighted.
>
> Below are the replies to your weakness concerns and questions.
>
> **Weaknesses 1:** For the node structure, this paper only compares CHT with fully connected graph, the author should take other graph structure into consideration, like ER, BA, WS graph.
>
> **Reply:** We express our sincere gratitude for the reviewer's insightful feedback. To address the reviewer’s concern, we have included a new section in Appendix L that delves into the selection of network initialization from a network science perspective.
> We clarify that the ER model initialization is the one already used in our study and this is explained now better in Section 3.1 Epitopological sparse meta-deep learning (ESML) and section 3.2 CH training strategy. Revisions are highlighted in red.
>
> Furthermore, we have designed procedures to generate Watts–Strogatz (WS) model and the Barabási-Albert (BA) models on bipartite networks and we considered the following scenarios: static sparse training with BA, WS and ER (equivalent to WS for beta = 1); dynamic sparse training again with BA, WS and ER (equivalent to WS for beta = 1); CHT with initialization of the network using BA, WS and ER (equivalent to WS for beta = 1). We clarify that in our previous version of the study, we initialized our network employing Correlated Sparse Topological Initialization (CSTI) on layers that directly interact with input features from the dataset and apply Erdos-Renyi (ER) to intermediate layers. Now, thanks to the Reviewer suggestions, we could investigate to initialize the sparse network structure also by using the Watts–Strogatz (WS) model and the Barabási-Albert (BA) model.
>
> The preliminary results that we gained on 3 datasets are consistent and show that in general the static sparse training via ER, BA or WS is outperformed by the respective dynamic sparse training that, in turn, is outperformed by CHT. In addition, in some cases, CHT initialized with BA and WS displays better performance than CHT initialized by ER or CSTI. We are expecting to obtain the results in the next few weeks on the other datasets and architectures, but we are quite confident that the trend obtained on the first 3 datasets is so consistent that might be confirmed also in the next datasets.
> We thank again the Reviewer for these valuable suggestions.
>
> **Weakness 2.** Compared with SOTA models, the performance of CHT is not competitive.
>
> **Reply:** We thank the Reviewer because it allows us to discuss this point that certainly needs further clarification in the manuscript. To address the Reviewer's concern, we are adding this new text in the section on Discussion of Limitations and future challenges which is provided in Appendix C.
>
> << With respect to the dynamic sparse training SOTA (RigL and MEST), the computational evidence in Fig. 3 on 5 empirical tests obtained in different combinations of network architectures/datasets (VGG16 on TinyImageNet; GoogLeNet on CIFAR100 and TinyImageNet; ResNet50 on CIFAR100 and ImageNet), demonstrate that CHT offers a remarkable ultra-sparse (1% connectivity) advantage on the fully connected baseline, whereas the current SOTA cannot. For the sake of clarity, we acknowledge that in this study we do not compare with the specific SOTA in the respective data or architecture types, because they often include methodologies with tailored technical features that are network and data-dependent. Including these specific SOTA, would make it difficult a fair comparison across data and networks because it would largely depend on the way we adapt the ultra-sparse training strategy to the specific case.  The attempt of this study is instead to evaluate whether, and in which cases, CHT can help to achieve ultra-sparse advantage in comparison to the fully connected baseline, regardless of the specific network or data considered>>

---

> ### Author Response · Authors · 2023-11-20
> **Reply to the Reviewer cMYk**
>
> **Questions:**
>
> 1. What's the inspiration of current architecture?
>
> **Reply:** We are grateful for this comment that we address in the Introduction section of the revised manuscript by introducing this paragraph:
>
> << The inspiration behind this theory is that for many networks associated to complex systems such as the brain, the shape of their connectivity is learned during a training process which is the result of the dynamic evolution of the complex systems across the time (ref). This means that the evolution of the complex system carves the network structure forming typical features of network complexity such as clustering, small-worldness, power-lawness, hyperbolic topology and community organization (ref). In turn, given a network with a shape characterized by these recognizable features of complexity, a significant (better than random) part of its future connectivity can be predicted by local network automata which interpret the information learned in the engrams (memory traces) of the complex network topology (ref). Based on these notions, network shape intelligence is the intelligence displayed by any topological network automata to perform valid (significantly more than random) connectivity predictions without training, by only processing the input knowledge associated to the local topological network organization (ref). The network automaton training is not necessary because it performs predictions extracting information from the network topology, which can be regarded as an associative memory trained directly from the connectivity dynamics of the complex system (ref). >>
>
> 2. Did you do extra experience on other graph structure?
>
> **Reply:** Please see the reply that we reported above to address the weakness 1.
>
> 3. In Table 2 on page15. Why some accuracy result is not the best but still highlighted?
>
> **Reply:** We are sorry if this was not clear in the previous version of the manuscript. To address the reviewer’s concern, we added the below content in all the captions of the Results Table.
>
> <<We executed all the DST methods in an ultra-sparse scenario (1%). The best performance among the sparse methods is highlighted in bold, and values marked with “*” indicate they surpass those of the fully connected counterparts.>>

---

> > ### Comment · Reviewer_cMYk · 2023-11-22
> >
> > Thank you for your prompt response and extra experiences. The authors addressed most of my concerns. I'll consider to increase the score.

---

### Official Review · Reviewer_M5bF · 2023-11-03

**Soundness:** 3 good
**Presentation:** 3 good
**Contribution:** 3 good
**Rating:** 8
**Confidence:** 2

**Summary:**

The authors propose a new form of sparse network learning, which was brain-inspired. They compare to other methods in the field and evaluate on vision dataset.

**Strengths:**

Originality: I am not very familiar with the literature in this community, but from what I can tell the authors cite existing literature well and distinguish their approach adequately from other methods in the field.
Quality: the paper is well written, and the empirical results are well done: they are thorough, include meaningful baselines, and cover a nice variety of datasets.
Clarity: the paper was well written
Significance: I am not familiar enough with this sub-field to judge, but given the result, this appears to be a solid step in the right direction, and is likely relevant to the subcommunity.

**Weaknesses:**

While I understand the space constraints, having something like Table1 or Table2 in the main text (maybe in reduced form) would be nice.

**Questions:**

No questions

---

> ### Author Response · Authors · 2023-11-20
> **Reply to the Reviewer M5bF**
>
> Dear Reviewer M5bF,
>
> Thank you for your insightful and constructive review of our manuscript. We greatly appreciate your recognition of its originality and quality.
>
> W1: While I understand the space constraints, having something like Table1 or Table2 in the main text (maybe in reduced form) would be nice.
>
> R1: Thanks for this value suggestion that we implemented by adding a reduced form of Table 1 directly below the Fig. 3, in order to simplify the reader in the visualizations and interpretations of the results.
>
> Best wishes,
>
> Authors of Submission 1203

---

> > ### Comment · Reviewer_M5bF · 2023-11-22
> >
> > Dear authors, thank you for taking the time to responding to my review. Best of luck with your submission!

---

### Author Response · Authors · 2023-11-20
**General REPLY and Updated Manuscript**

After submitting the article, we collected more evidence that CHT outperforms the dense network with only 1% links remaining by adding tests on new network models and datasets. Please refer to the updated Figure 3 and Table 1 in the revised manuscript for these enhancements.

Additionally, we are thankful to:
+ The first Reviewer for advising on how to improve the presentation of the main results.
+ The second Reviewer for suggesting a thoughtful approach to network topology initialization, an aspect we had not considered in our initial submission. We have included a discussion on this in Appendix L, further enriching the completeness of our article.
+ The third Reviewer for the valuable comments on considering the impact of reduced depth and width in models applied to large datasets. To address it, we considered larger and more difficult datasets such as TinyImageNet instead of CIFAR10 on VGG16, for a more direct comparison with a new considered CNN model GoogLeNet. Finally, we also performed a test on CHT performance in which we reduced the intermediate network size from 4 to 2 layers on a relatively large dataset whose computation could be finalized before the rebuttal deadline.

We already updated the manuscript in many parts, but due to time limitations, some other parts will need to be adjusted before camera-ready if the study is accepted.

---

### Meta-Review · Area_Chair_tYFU · 2023-12-10

**Metareview:**

This paper implements ultra-sparse network topologies (~1% synapses) using Cannistraci-Hebb training, which (roughly speaking) is a procedure to forecast the local structure of the synapses as the network crystallizes the connectivity between the synapses during the course of learning. The approach has strong connections to meta-learning and multi-step gradient descent-based methods. The paper shows that this approach improves the speed and accuracy of training as compared to MLPs on MNIST and CNNs on CIFAR-10/100 and TinyImagenet, and uses only 1% of the weights.

**Justification For Why Not Higher Score:**

These results are impressive but preliminary, e.g., the accuracy on MNIST is ~80% and CIFAR-100 is ~60%. These results are achieved (esp. the latter) with only 1% of the weights but using a very complicated training procedure. Current methods for sparse training/pruning top out at ~10% of the number of weights.

**Justification For Why Not Lower Score:**

The technical approach presented in the paper is very complicated but interesting. It is worthwhile for the research community to get exposure to these ideas with the hope that these techniques can be simplified in the future and marshaled better.

---

### Decision · Program_Chairs · 2024-01-16

Accept (poster)